# Nec-1 alleviates cognitive impairment with reduction of Aβ and tau abnormalities in APP/PS1 mice

Seung-Hoon Yang[1], Dongkeun Kenneth Lee[1], Jisu Shin[1], Sejin Lee[1,2], Seungyeop Baek[1,3], Jiyoon Kim[1,2], Hoyong Jung[4], Jung-Mi Hah[4] & YoungSoo Kim[1,2,*]

## Abstract

Alzheimer's disease (AD) is a neurodegenerative disorder characterized by cognitive symptoms of learning and memory deficits. Such cognitive impairments are attributed to brain atrophy resulting from progressive neuronal and synaptic loss; therefore, alleviation of neural cell death is as an important target of treatment as other classical hallmarks of AD, such as aggregation of amyloid-β (Aβ) and hyperphosphorylation of tau. Here, we found that an anti-necroptotic molecule necrostatin-1 (Nec-1) directly targets Aβ and tau proteins, alleviates brain cell death and ameliorates cognitive impairment in AD models. In the cortex and hippocampus of APP/PS1 double-transgenic mice, Nec-1 treatment reduced the levels of Aβ oligomers, plaques and hyperphosphorylated tau without affecting production of Aβ, while it altered the levels of apoptotic marker proteins. Our results showing multiple beneficial modes of action of Nec-1 against AD provide evidence that Nec-1 may serve an important role in the development of preventive approach for AD.

**Keywords** Alzheimer's disease; Aβ aggregation; cognitive deficit; necrostatin-1; tau hyperphosphorylation
**Subject Categories** Neuroscience; Pharmacology & Drug Discovery

## Introduction

Neural and synaptic loss, indicated by gradual cerebral atrophy, is a prominent feature of Alzheimer's disease (AD) (Terry *et al*, 1991; Jack *et al*, 2000, 2004). Neural cell death especially in the cortex and hippocampus is responsible for the decline of learning and memory abilities observed in AD patients (Double *et al*, 1996; Niikura *et al*, 2006). These brain lesions are accompanied by the accumulation of senile plaques and neurofibrillary tangles, which mainly consist of aggregated amyloid-β (Aβ) and hyperphosphorylated tau, respectively (Schoonenboom *et al*, 2004; Sobow *et al*, 2004).

In developing drugs for AD, different approaches have been taken in many extensive investigations; however, most drugs currently available serve to modulate the degree of symptoms without affecting the fundamental causes (Yiannopoulou & Papageorgiou, 2013; Kim *et al*, 2015). Thus, drug candidates targeting the prevention of neural loss as well as the alleviation of Aβ aggregation and tau hyperphosphorylation would be promising therapeutic approach to treat AD (Hong-Qi *et al*, 2012; Mondragon-Rodriguez *et al*, 2012).

As neurodegeneration is highly associated with AD progression (Serrano-Pozo *et al*, 2011), it is essential to investigate a means to regulate cell death in the brain in order to find therapeutic interventions for AD. The small molecule necrostatin-1 (Nec-1) is a well-established inhibitor of necroptotic cell death, also known as necroptosis (Degterev *et al*, 2005). In particular, Nec-1 is reported to have neuroprotective effects in several brain disease models by reducing tissue damage in brain injury, alleviating functional deficits in Huntington's disease and inhibiting neural cell degeneration induced by aluminium (Zhu *et al*, 2011; Wang *et al*, 2012; Qinli *et al*, 2013; Su *et al*, 2015). Many studies have indicated that necroptosis inhibition by Nec-1 can be attributed to its regulation of a signalling complex containing members of the receptor interacting protein kinase (RIPK) family, RIPK1 and RIPK3 (Degterev *et al*, 2008; Vandenabeele *et al*, 2013). Recently, the RIPK1/RIPK3 signalling complex has been reported to exhibit amyloidal properties required for necroptosis (Li *et al*, 2012).

Taking all of these findings from previous studies into consideration, we hypothesized that Nec-1 could modulate neural cell death and that its effects could possibly extend to treating amyloidal properties of Aβ and tau in the AD brain. We aimed to search for prophylactic treatment targeting multiple neuropathological changes before outward symptoms of AD become apparent because aggregation of Aβ and hyperphosphorylation of tau begin in the pre-symptomatic

1  Convergence Research Center for Dementia and Center for Neuro-Medicine, Brain Science Institute, Korea Institute of Science and Technology, Seoul, Korea
2  Biological Chemistry Program, Korea University of Science and Technology, Daejeon, Korea
3  Department of Biotechnology, Yonsei University, Seoul, Korea
4  Department of Pharmacy, College of Pharmacy & Institute of Pharmaceutical Science and Technology, Hanyang University, Ansan, Kyeonggi-do, Korea
  *Corresponding author. Tel: +82 2 958 5161; E-mail: yskim@bio.kist.re.kr

stage that intervening AD as early as possible is deemed more effective and practical (Jack *et al*, 2010, 2013). In this study, we investigated the effects of Nec-1 on Aβ-induced neural cell death *in vitro* using multiple cell models. Furthermore, APP/PS1 double-transgenic mice were subjected to behavioural tests to evaluate whether Nec-1 alters cortical- and hippocampal-dependent cognitive functions, and the brains were examined for changes in the levels of Aβ plaques, oligomers, hyperphosphorylated tau and apoptotic marker proteins. Additionally, bimolecular interactions of Nec-1 with Aβ or tau were studied to further understand the effects of Nec-1 in relation to AD aetiology.

## Results

### Nec-1 blocks Aβ-induced neural cell death

The transition of Aβ monomers into neurotoxic aggregates serves as a pathological trigger in AD, ultimately resulting in cerebral atrophy (Irvine *et al*, 2008). To investigate the effect of Nec-1 on Aβ-induced brain cell death, we first examined the cytotoxicity of Nec-1 in neural cells. In HT22 cells (hippocampal neuronal cell line) treated with different concentrations of Nec-1, we did not observe decreases in cell viability while Aβ(1–42) aggregates induced neural cell death (Fig 1B). Then, we conducted cell viability assays using BV2 (microglial cell line) and primary cultured astrocytes derived from the cortex and hippocampus of C57BL/6 mouse brains in addition to HT22 cells. Nec-1 (50 μM) was applied to each cell model (pre-treatment) 15 min before Aβ(1–42) aggregates (10 μM) were added to cells. We prepared Aβ(1–42) aggregates by incubating Aβ(1–42) monomers (1 mM) at 37°C overnight, then diluting the peptide stock with cell culture medium. Cells cultured in the absence of Nec-1 and Aβ(1–42) aggregates were used as controls (non-treated), and the level of cell viability was quantified by MTT assays. Cell proliferation significantly decreased in all cultured cells treated with Aβ(1–42) aggregates compared to non-treated cells (Fig 1C). However, for all three types of cells with Nec-1 pre-treatment, the cell viabilities did not significantly decrease in the presence of Aβ(1–42) aggregates, indicating that Nec-1 can effectively block the cytotoxicity of Aβ(1–42) (Fig 1C). To further confirm the anti-cytotoxic effect of Nec-1, we used the LIVE/DEAD® Viability/Cytotoxicity Assay. In this assay, live cells are distinguished from dead cells

by two different fluorescent molecules: calcein AM indicating live cells (green colour) and ethidium homodimer-1 (EthD-1) indicating dead cells (red colour) (Xie *et al*, 2009). Although non-treated cells showed some endogenous cell death, this was severely increased after the addition of Aβ(1–42) aggregates. However, the levels of cell death were not significantly different between Nec-1 pre-treated cells and non-treated cells (Fig 1D and E). These results indicated that Nec-1 can block Aβ(1–42)-induced neural cell death *in vitro*.

### Nec-1 regulates Aβ-induced neural cell death by blocking the apoptotic pathway

Extracellular Aβ is known to be internalized into neural cells, and such internalization is associated with cell death in the brain (Friedrich *et al*, 2010; Kim *et al*, 2013b). To determine whether Nec-1 affects Aβ penetration into neural cells, Aβ(1–42) levels in the cytoplasmic fractions of neuronal and microglial cells were analysed by Western blot. HT22 and BV2 cell lines were pre-treated with Nec-1 15 min prior to the addition of Aβ(1–42) (10 μM). We observed that Nec-1 treatment did not affect the level of Aβ(1–42) monomers in the cytoplasm of either HT22 or BV2 cell lines (Fig 1F). This result indicated that Aβ(1–42) penetration into the cell is not affected by Nec-1.

To explore the cell death signalling pathway by which Nec-1 blocks Aβ-induced neural cell death, expression levels of apoptotic molecules were measured in HT22, BV2 and primary cultured astrocyte lysates by Western blot. The molecules investigated were poly (ADP-ribose) polymerase-1 (PARP1), which is specifically proteolysed by caspases in response to genotoxic stress, such as DNA damage, to induce apoptotic cell death (D'Amours *et al*, 2001); cleaved caspase-3, which plays a central role in the apoptotic signalling pathway (Salvesen, 2002); cytochrome-c, a small molecule released from the mitochondria into the cytoplasm during apoptosis (Liu *et al*, 1996); Bax, a pro-apoptotic member of the Bcl-2 protein family (Oltvai *et al*, 1993; Lalier *et al*, 2007); Bcl-2, an anti-apoptotic protein that inhibits activation of the apoptotic cell death pathway (Belka & Budach, 2002); and phosphorylated c-Jun N-terminal kinase (JNK), which activates apoptotic signalling by upregulating pro-apoptotic genes (Dhanasekaran & Reddy, 2008). Each cell model was pre-treated with Nec-1 before cell death was induced by Aβ(1–42) aggregates (10 μM). In BV2 cell lysates, Nec-1 pre-treatment led to marked inhibition in the Aβ(1–42)-induced processing rates of

---

**Figure 1. Nec-1 inhibits neural cell death in response to Aβ aggregates.**

A    Chemical structure of Nec-1. The schematic diagram was drawn using ChemDraw Professional 15.0. software.

B, C    MTT cytotoxicity assay. (B) HT22 cell line was treated with various concentrations of Nec-1 (0–200 μM). Aggregated Aβ(1–42) was used as the positive control for the induction of neural cell death. (C) HT22, BV2 cell lines and primary astrocytes were pre-treated with Nec-1 (50 μM) 15 min prior to application of 10 μM of pre-aggregated Aβ(1–42) for indicated times. Cell proliferation was then measured by MTT assay. All data are representative results of three independent experiments performed in triplicate.

D, E    Neural cell death levels measured by staining with the LIVE/DEAD Viability/Cytotoxicity Assay Kit (Molecular Probes). Pre-treatment of cells with 50 μM Nec-1 preceded the application of 10 μM pre-aggregated Aβ(1–42) by 15 min. The assay was performed 24 h after Aβ(1–42) application. (D) Representation of the stained cells. Scale bars = 500 μm. (E) Quantification of dead cells stained by EthD-1. All data are representative results of at least three independent experiments.

F    Internalization of Aβ(1–42) into HT22 and BV2 cell lines in the presence or the absence of Nec-1 pre-treatment for the indicated periods.

G    Cellular expression levels of indicated proteins in HT22, BV2 cell lines and primary astrocyte stimulated with Aβ(1–42) aggregates in the presence or the absence of Nec-1 pre-treatment for 24 h. Low, low-exposure of blot to film; High, high-exposure of blot to film.

Data information: In (B, C and E), data are presented as mean ± SEM. *$P \leq 0.05$, **$P \leq 0.01$ and ***$P \leq 0.001$ (one-way ANOVA followed by Bonferroni's *post hoc* comparisons tests). Exact *P*-values are shown in Table EV1. NT, non-treated cells.

Source data are available online for this figure.

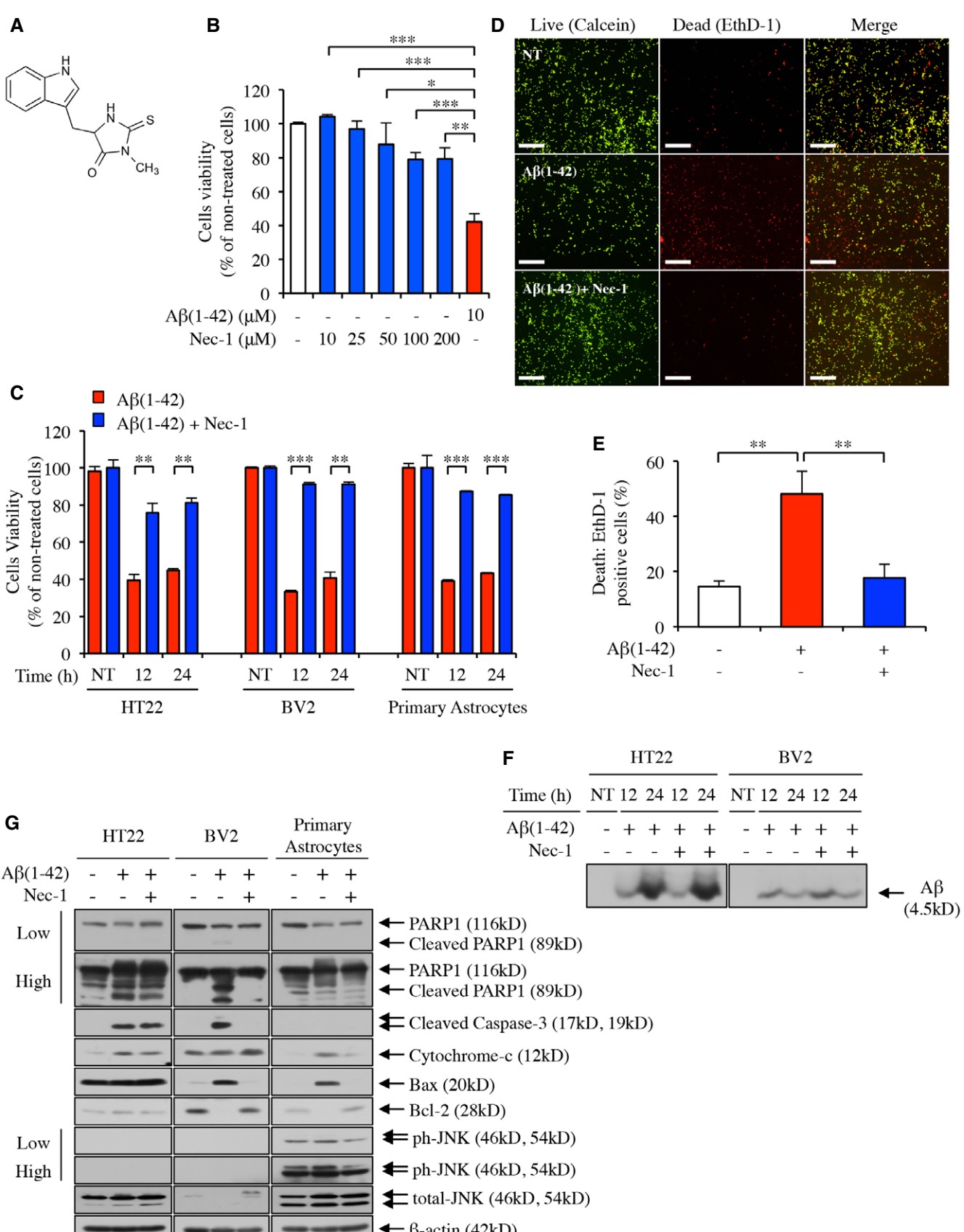

Figure 1.

PARP1, caspase-3 and Bax. In addition, Bcl-2 expression was increased, whereas no significant difference was observed in cytochrome-c expression (Fig 1G). In primary astrocytes, marker protein expression levels followed a similar trend to that observed in BV2 cells. The pro-apoptotic markers cytochrome-c and Bax were highly expressed in the presence of Aβ(1–42) aggregates, but their levels decreased with Nec-1 pre-treatment. The expression of anti-apoptotic Bcl-2, which was hardly observed in the presence of Aβ(1–42) aggregates, dramatically increased with Nec-1 pre-treatment. Furthermore, the level of phosphorylated JNK was reduced in primary astrocytes pre-treated with Nec-1, indicating decreased apoptotic cell death (Fig 1G). However, HT22 cells exhibited no significant Aβ-induced differences in apoptotic marker protein levels, except for cytochrome-c expression, which was slightly decreased by Nec-1 pre-treatment (Fig 1G). Taken together, these findings suggested a role of Nec-1 in blocking neural cell death via an apoptotic pathway induced by Aβ.

### Nec-1 prevents deficits in learning and memory abilities in AD-like mouse models

The inhibitory effects of Nec-1 on Aβ-induced cytotoxicity suggest a potential application of the molecule to treat Aβ-related learning and memory deficits. To assess the prophylactic effect of Nec-1 on spatial working and hippocampal memory, we utilized the APPswe/PS1-dE9 double-transgenic mouse model (APP/PS1) (Jankowsky et al, 2004). This mouse model expresses mutant human amyloid precursor protein (APP) and presenilin protein 1 (PS1), causing AD-like amyloid plaque formation as early as 4 months of age while cognitive impairment becomes observable around 6–8 months of age (Savonenko et al, 2005; Jackson et al, 2013). To 4-month-old male APP/PS1 mice, we intravenously injected Nec-1 (n = 7) or vehicle (n = 6) for 12 weeks until the age of 7 months (Fig 2A). As controls, we intravenously injected age-matched male wild-type (Wt) mice with Nec-1 (n = 7) or vehicle (n = 7) for 12 weeks. We subjected these mice to Y-maze and passive avoidance tests 2 days after the last administration of Nec-1 to assess alterations in learning and memory abilities.

In the three-armed Y-shaped maze, a mouse is free to visit the three arms in any order, and the number of entries to the three different arms in a row is measured and compared to the number of total visits. The resulting per cent alternation can be used as a measure for spatial working memory, as a mouse with higher visual cortex function demonstrates higher per cent alternation (Gotz &

Ittner, 2008). In this study, we found that Nec-1 administration significantly improved behavioural performance of the APP/PS1 mice compared to that of the vehicle-administered APP/PS1 group (P = 0.01465), as suggested by increased per cent alternation. The spatial working memory of APP/PS1 mice was recovered up to Wt levels after Nec-1 administration (Fig 2B). In addition, we did not observe behavioural differences between Nec-1 and vehicle-administered Wt groups. No significant difference in total number of arm entries was found, excluding hyperactivity as a possible confounding factor for cognitive improvement (Fig 2C).

To evaluate the effect of Nec-1 on hippocampal-dependent memory in APP/PS1 mice, we performed the passive avoidance test. Mice were rested for 3 days after completion of the Y-maze test, and no additional injections were provided (Fig 2A). In this test, the latency of movement from a bright to a dark chamber is measured and compared before and after an electrical shock application. A difference in latency following an aversive experience (retention) reflects contextual memory recollection, and shorter latencies indicate impairment in hippocampal memory function (Lorenzini et al, 1996). We did not find any significant difference in latencies between Wt and APP/PS1 mice, regardless of Nec-1 administration, before shock application (acquisition). In the retention trial, however, APP/PS1 mice showed significantly shorter latency compared to Wt mice (P = 0.00362) when both groups were administered with vehicle, whereas APP/PS1 mice administered with Nec-1 showed greatly enhanced latency compared to APP/PS1 mice administered with vehicle (P = 0.0429) (Fig 2D). All mice we examined survived throughout the aforementioned behavioural studies, indicating that Nec-1 administration does not have lethal effect in both Wt and APP/PS1 mice (Fig 2E). These results suggest that Nec-1 administration in APP/PS1 mice prevented cognitive impairment related to the Aβ cascade.

To confirm whether Nec-1 administration could also affect AD-like cognitive impairment of a different mouse model, we acutely induced memory deficits in Imprinting Control Region (ICR) mice (8-week-old male, n = 8 per group) by intracerebroventricular injection of Aβ(1–42) aggregates (Fig 2F and G). The benefit of this Aβ(1–42) infusion model is that we can control the onset of abnormal Aβ deposition as well as timely treatment of Nec-1 (Kim et al, 2016). To examine the prophylactic effect of Nec-1 in the Aβ(1–42) infusion model, Nec-1 was intravenously injected every 3 days for a week, followed by an intracerebroventricular injection of Aβ(1–42) aggregates and additional two injections of Nec-1 for another week. We conducted Y-maze tests 2 days after the last injection of Nec-1

---

**Figure 2. Nec-1 inhibits spatial working memory and hippocampal memory declines in AD-like mouse models.**

A–E  Behavioural tests of APP/PS1 mice. (A) Schedule of behavioural tests. Four-month-old male APP/PS1 mice were intravenously injected with Nec-1 (6.25 mg/kg, n = 7) or vehicle (2.5% DMSO in PBS, n = 6) for 12 weeks (2 times per week). As controls, age-matched male wild-type (Wt) mice were injected with Nec-1 (6.25 mg/kg, n = 7) or vehicle (2.5% DMSO in PBS, n = 7) for 12 weeks. (B–D) Y-maze and passive avoidance tests on 7-month-old APP/PS1 mice after Nec-1 administration for 12 weeks. Average alternation (%) for each test group (B) and total entry number into each arm (C) on Y-maze. (D) Average latency time in seconds for each test group on passive avoidance test. (E) Survival of Wt and APP/PS1 mice after injection of Nec-1 (6.25 mg/kg).

F–I  Behavioural tests of Aβ(1–42) infusion mice. (F) Schedule of behavioural tests. In the first week, 8-week-old male ICR mice (n = 8 per each group) were intravenously injected with either Nec-1 (6.25 mg/kg) or vehicle (2.5% DMSO in PBS) twice. Then, 5 μl of Aβ(1–42) (100 μM) aggregates or vehicle (10% DMSO in PBS) were injected in intracerebroventricular region to generate Aβ(1–42) infusion mice model. In second week, same concentrations of Nec-1 or vehicle were intravenously injected twice again. (G) Intracerebroventricular (i.c.v.) injection site brain schematic diagram. (H, I) Y-maze tests on Aβ(1–42) infusion mice model injected with Nec-1. Average alternation (%) for each test group (H) and total entry number into each arm (I) on Y-maze.

Data information: In (B–D, H and I), data are presented as mean ± SEM. *P ≤ 0.05, **P ≤ 0.01 and ***P ≤ 0.001 (one-way ANOVA followed by Bonferroni's post hoc comparisons tests). Exact P-values are shown in Table EV2.

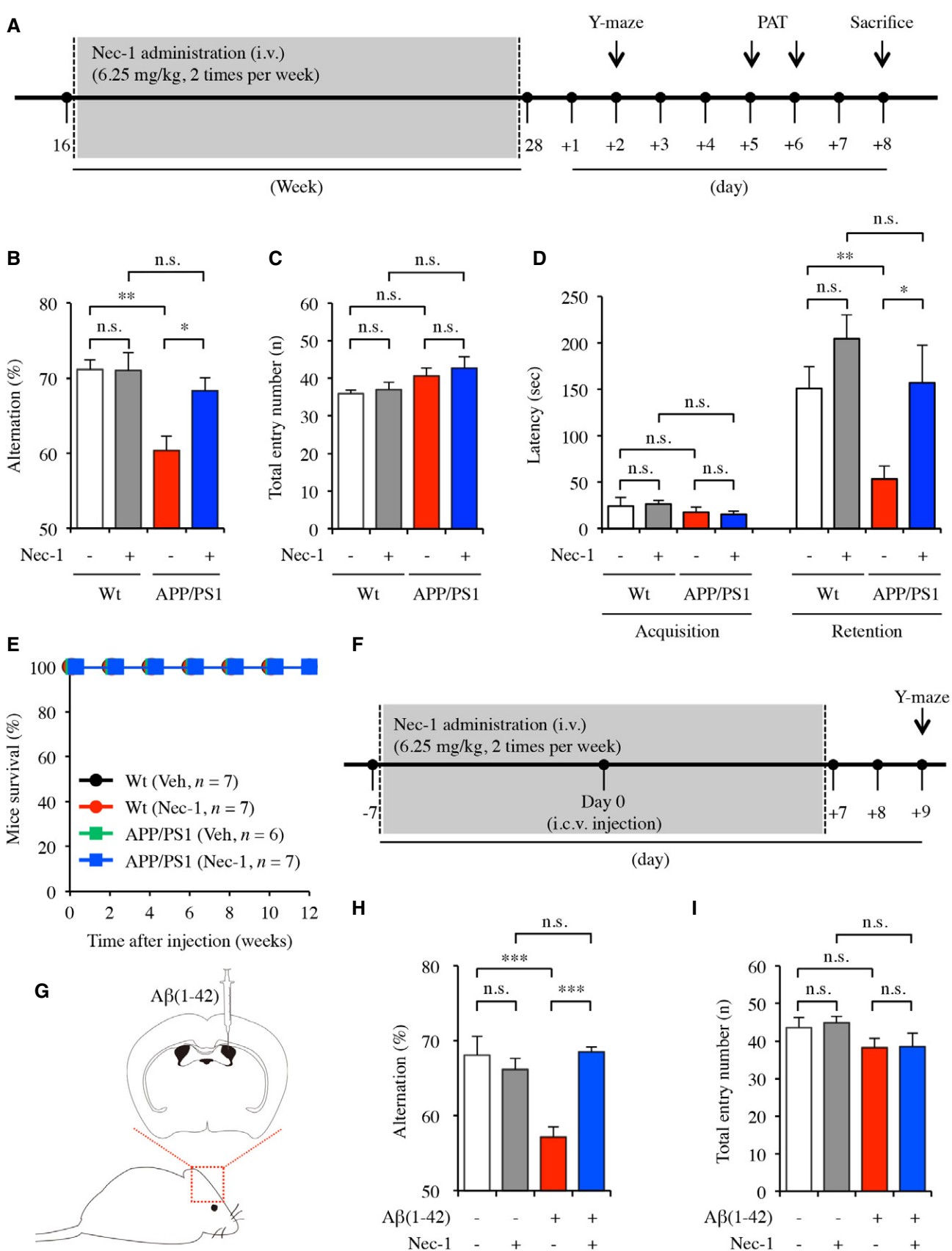

Figure 2.

to assess alterations of short-term spatial working memory (Fig 2F). In the Y-maze test, Nec-1 injections significantly ameliorated Aβ(1–42)-induced memory deficits ($P = 0.0002$) (Fig 2H and I). These results imply that Nec-1 prevented Aβ-induced cognitive impairments in the Aβ(1–42) infusion model as well.

### Nec-1 significantly reduces Aβ plaques and oligomers in the cortex and hippocampus of APP/PS1 mice

Aβ aggregates into insoluble clusters, or plaques, in the AD brain as the disease progresses (Naslund *et al*, 1994; Shinkai *et al*, 1997). Formation of such plaques in the cortical and hippocampal regions is directly related to learning and memory deficits in AD (Jahn, 2013). Given that Nec-1 targets the heterodimeric filamentous complex of RIPK1/RIPK3 (Cho *et al*, 2011), which is similar to the characteristic of Aβ misfolding (Li *et al*, 2012), we hypothesized that Nec-1 might directly bind to Aβ aggregates. First, we explored the interaction between Aβ(1–42) and RIPK3, a signalling molecule involved in Nec-1-regulated cell death (Degterev *et al*, 2008; Vandenabeele *et al*, 2013). HT22, BV2 and primary astrocytes were treated with Aβ(1–42) (10 μM). At 12 and 24 h following Aβ treatment, cell lysates were immunoprecipitated with anti-RIPK3 antibody and subjected to Western blot analyses to measure Aβ(1–42) levels. Using a monoclonal anti-Aβ antibody (6E10), we observed the presence of Aβ(1–42) bound to RIPK3, suggesting a similarity in amyloidal properties between RIPK3 and Aβ(1–42) (Fig 3A). This result raised the possibility that Nec-1 may directly bind to Aβ aggregates. According to a previous report on the crystal structure of RIPK1 in complex with Nec-1 analog, Nec-1a, it was caged in a hydrophobic pocket between the N- and C-lobes of the kinase domain, having mostly van der Waals interactions with hydrophobic residues (L70, V75, V76, L78, L90, L129, V134, L157, L159, F162) and a few hydrogen bondings (D156, S161) (Xie *et al*, 2013). As we observed that Nec-1 also binds to the same hydrophobic pockets of RIPK1 (Fig 3B), we searched by modelling for a similar sized-hydrophobic site, where Nec-1 could bind in Aβ aggregates. When the available structure of 12-mer Aβ fibrils was examined as surface treatment (Fig 3C), we observed several plausible hydrophobic pockets, within size of 9–14.5 Å, and Nec-1 consequently docked in those pockets. Without induced fit, we could simulate four molecules of Nec-1 docked on the Aβ fibrils with substantial docking scores (−8.338, −4.755, −8.493 and −6.754 in clockwise order starting from the top-left corner) and ligand efficiency (Fig 3D). To confirm a possible interaction between Nec-1 and Aβ aggregates, we applied a label-free surface plasmon resonance assay

to estimate bimolecular binding kinetics using the Biacore T200 system. After immobilization of Aβ(1–42) aggregates on CM5 dextran matrix sensor chips via amine coupling, Nec-1 was injected into the microfluidic channel as an analyte. Kinetic analysis was performed at a concentration range of 0.038–39.06 nM. Although the precise binding constant could not be calculated due to the heterogeneous nature of Aβ aggregates and it is uncommon to measure bimolecular interactions between a small molecule and heterogeneous misfolded proteins, the direct and dose-dependent interaction of Nec-1 with immobilized Aβ aggregates was indicated by the substantial curve fitting efficiency (6.18 $RU^2$ of calculated $\chi^2$) (Fig 3E).

To examine the effect of Nec-1 on plaque formation in the brain, we observed the levels of Aβ plaques in mice subjected to the aforementioned behavioural studies. Mice were anesthetized with 2% avertin (20 μg/g, i.p.) and perfused with 0.9% NaCl. Extracted brains were fixed with 4% paraformaldehyde and immersed in 30% sucrose for cryoprotection. Fixed brain samples were sectioned at 30 μm and stained with thioflavin S (ThS) to visualize insoluble β-sheet-rich Aβ plaques. Total numbers and areas of plaques found in the whole brain, cortex or hippocampus were quantified. In the cortex and hippocampus of APP/PS1 mice, we observed numerous plaques at 7 months of age, but plaque formation was dramatically inhibited in Nec-1-administered APP/PS1 mice (Fig 4A). Total numbers and areas of plaques in whole brains of Nec-1-administered APP/PS1 mice were notably decreased compared to those with the vehicle administration (Fig 4B and C). This trend was consistent in both the cortical (Fig 4D and E) and hippocampal (Fig 4F and G) regions, implying that Nec-1 administration affects plaque formation in different brain regions.

Aβ plaques can be classified into dense-core and diffuse plaques. Dense-core plaques differ from diffuse plaques in that only dense-core plaques can be stained with ThS due to their aggregated β-sheet structure, which is not present in diffuse plaques (Rak *et al*, 2007). Diffuse Aβ plaques in AD have been suggested to be associated with reactive astrocytes that cluster around them and result in neurodegenerative events (Funato *et al*, 1998; Yamaguchi *et al*, 1998). Therefore, we performed immunohistochemical staining in the brains of mice that underwent behavioural studies to detect diffuse plaques and reactive astrocytes, using the 6E10 antibody and anti-glial fibrillary acidic protein (GFAP) antibody, respectively. We found that diffuse plaques co-localized with reactive astrocytes in APP/PS1 mouse brains, and that Nec-1 administration induced decreases in the levels of diffuse plaques as well as GFAP in the cortex and hippocampus of APP/PS1 mice (Fig 5A).

---

**Figure 3.  Nec-1 binds to Aβ aggregates.**

A   Co-immunoprecipitation of RIPK3 with Aβ(1–42) in HT22, BV2 cell lines and primary astrocytes in response to Aβ(1–42) aggregates. IP, immunoprecipitation; IB, immunoblot.

B   A close-up view of the interactions between Nec-1 and surrounding residues in RIPK1. Nec-1 is shown in green. The hydrophobic residues around Nec-1 in RIPK1 kinase domain are shown in sticks. Hydrogen bonds are represented by yellow dashed lines. Docking score of Nec-1 in RIPK1 was −9.550.

C   Surface treated view of 12-mer Aβ fibrils (2LMO, VDW radius = 0.55 Å) having hydrophobic pockets plausible for Nec-1 binding.

D   Docked structure of 12-mer Aβ fibrils (2LMO) with four molecules of Nec-1. Docking scores for four top-ranked Glide docking poses of Nec-1 in Aβ plaques: −8.338, −4.755, −8.493 and −6.754 in clockwise order starting from the top-left corner. Hydrogen bonds are represented by red dashed lines.

E   Surface plasmon resonance sensorgrams of Nec-1 targeting Aβ(1–42) aggregates (left), and the corresponding dissociation fitting curve from the saturated region (right) under various concentrations of Nec-1.

Source data are available online for this figure.

**Figure 3.**

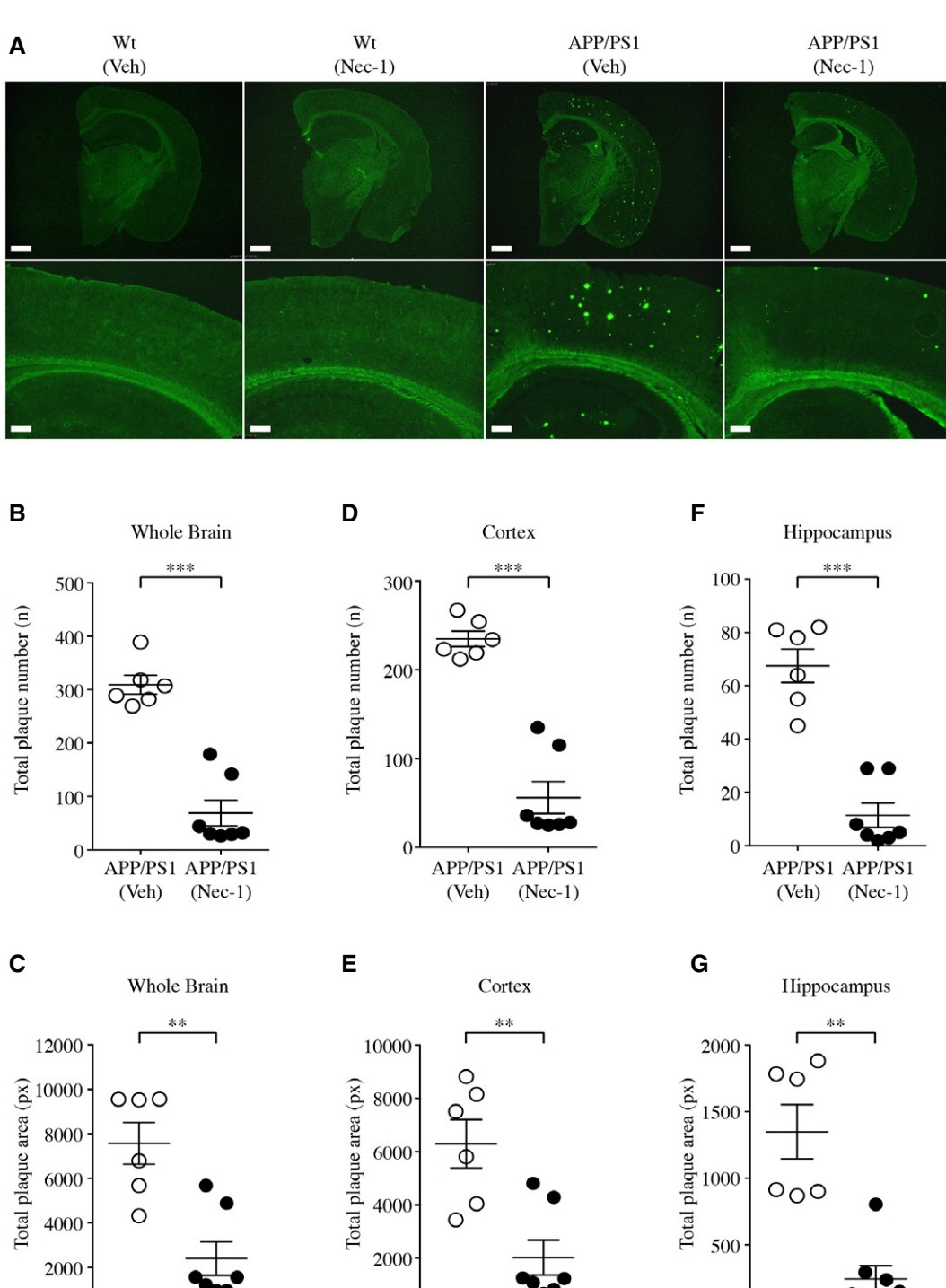

**Figure 4. Nec-1 reduces insoluble Aβ plaques in the brains of APP/PS1 mice.**

Four-month-old APP/PS1 (*n* = 13, male) and age-matched wild-type (Wt, *n* = 14, male) mice were intravenously injected with Nec-1 (6.25 mg/kg) or vehicle (2.5% DMSO in PBS) for 12 weeks. After the completion of behavioural tests, the brains of each group of mice were subjected to analysis.

A ThS-stained insoluble Aβ plaques in whole brain and the hippocampal region of each group. Scale bars = 1 mm (upper), 200 μm (lower).

B–G Statistics of ThS-positive Aβ plaques in the brains of APP/PS1 mice. Total numbers and areas of ThS-positive Aβ plaques in the whole brains (B and C), cortex (D and E) and hippocampus (F and G).

Data information: In (B–G), data are presented as mean ± SEM. **$P \leq 0.01$ and ***$P \leq 0.001$ (one-way ANOVA followed by Bonferroni's *post hoc* comparisons tests). Exact *P*-values are shown in Table EV3.

In the AD brain, Aβ oligomers are a major cause of neural and synaptic loss due to their high toxicity (Sakono & Zako, 2010; Benilova *et al*, 2012). To investigate whether Nec-1 can prevent the formation of Aβ oligomers, we performed dot blot analyses using 6E10 and anti-amyloidogenic protein oligomer (A11) antibodies in cortex and hippocampus lysates from the mice in the aforementioned tests. 6E10 detects all Aβ species and also recognizes APP, whereas A11 only detects oligomeric proteins. We found that Nec-1 decreased the levels of oligomeric species, but the concentration of total soluble Aβ, which included monomers, was not significantly altered (Fig 5B–D). Furthermore, we measured soluble and insoluble Aβ(1–42), which were separated from cortex and hippocampus of the mice in the aforementioned tests by RIPA (soluble fraction) and guanidine (insoluble fraction) lysis buffers, by enzyme-linked immunosorbent assay (ELISA). As a result, we found that both soluble and insoluble Aβ(1–42) levels were decreased in the brains of APP/PS1 mice injected with Nec-1 (Fig 5E). Western blot analysis showed that Nec-1 did not affect the levels of APP and sAPPα in the cortex and hippocampus of APP/PS1 mice, suggesting that Nec-1 does not affect Aβ production (Fig 5F).

Next, we examined whether Nec-1 has a direct effect on Aβ oligomerization by the thioflavin-T (ThT) assay. ThT is known to detect beta-sheet-rich protein aggregates (Wolfe *et al*, 2010). We incubated monomeric Aβ(1–42) for 5 days to allow for the aggregation. When Aβ(1–42) was incubated in the presence of Nec-1, we observed that Aβ(1–42) aggregation was dramatically decreased, suggesting that Nec-1 has an inhibitory effect on Aβ aggregation (Fig 5G). To further visualize the effect of Nec-1 against Aβ aggregation, we performed SDS–PAGE with photo-induced cross-linking of the unmodified proteins (PICUP) followed by silver staining to monitor Aβ oligomerization and fibrillization (Bitan, 2006). We found that Nec-1 strikingly reduced both Aβ oligomers and fibrils (Fig 5H). Taken together, Nec-1 directly binds to Aβ aggregates and prevents the formation of Aβ plaques and oligomers in the brains of APP/PS1 mice.

### Nec-1 inhibits the phosphorylation and aggregation of tau

As hyperphosphorylation and aggregation of tau proteins are highly associated with brain cell death in AD, changes in tau abnormalities are important indications of AD progression (Calissano *et al*, 2009; Amadoro *et al*, 2011). To assess the effects of Nec-1 on tauopathy in

AD, we measured alterations in the levels of total and phosphorylated tau with apoptotic markers in the brains of mice previously studied in behavioural tests. We used a monoclonal antibody against phosphorylated-S199 (Catalog ab81268, Abcam), which is an important post-translation modification site of tauopathy and shared by both human and murine tau proteins. Nec-1 reduced the levels of phosphorylated tau, phospho-S199, in the cortex and hippocampus of APP/PS1 mice (Fig 6A). Consistent with our cell-based studies, the cortex of APP/PS1 mice after Nec-1 injections showed increased expression of the anti-apoptotic protein Bcl-2 as well as slightly decreased expression of pro-apoptotic proteins, including cytochrome-c and cleaved caspase-3, compared to the vehicle-administered group (Fig 6A). To further confirm the effect of Nec-1 on inhibition of tau phosphorylation, we conducted immunohistochemical analysis of phosphorylated S199 of tau in brains of aforementioned APP/PS1 mice. Nec-1 administration dramatically reduced levels of phosphorylated tau, specifically the phospho-S199 site, in both cortical and hippocampal regions of APP/PS1 mice (Fig 6B).

Next, we assessed the inhibitory effect of Nec-1 on necroptotic signalling in the brain of APP/PS1 mice. Given that Nec-1 regulates a signalling complex containing members of RIPK family (Degterev *et al*, 2008; Vandenabeele *et al*, 2013), we measured phosphorylation levels of RIPK3 by Western blot analysis. In APP/PS1 mice, Nec-1 reduced the phosphorylation level of RIPK3 in both cortex and hippocampus. Further analysis of primary cultured astrocytes showed the consistent result of reduced phosphorylation level of RIPK3 by Nec-1 (Fig 6C).

We examined whether Nec-1 has a direct effect on tau aggregation by the thioflavin-T (ThT) assay. In this study, we used the recombinant K18 fragment of full-length human tau, as it is an important binding domain involved in tau aggregation (Haque *et al*, 2014). First, we incubated a mixture of the K18 fragment and heparin in the presence of Nec-1 for 5 days. We found that Nec-1 dramatically inhibited aggregation, suggesting that Nec-1 has an inhibitory effect on tau aggregation (Fig 6D). Moreover, ThT fluorescence intensity was greatly reduced when Nec-1 was added to preformed K18 aggregates, implying that Nec-1 can also reduce tau complexes (Fig 6E). To understand how Nec-1 interacts with tau, we investigated the bimolecular binding kinetics between Nec-1 and tau by label-free surface plasmon resonance assay using the Biacore T200 system. We immobilized K18 on CM5 dextran matrix sensor chips

---

**Figure 5.  Nec-1 decreases diffuse plaques and Aβ oligomers in the brains of APP/PS1 mice.**

A Immunohistochemical analysis of cortical and hippocampal regions of the brains in wild-type (Wt) and APP/PS1 mice after administration of Nec-1 (6.25 mg/kg) or vehicle (2.5% DMSO in PBS). Diffuse plaques in the brain sections were stained by anti-Aβ antibody (clone 6E10, green colour) and anti-GFAP antibody (red colour). Hoechst 33342 (blue colour) was applied for nuclear counterstaining. Scale bars = 200 μm.

B–D Dot blot analysis of total Aβ (anti-Aβ: clone 6E10, also recognizes APP) and protein oligomer (anti-amyloidogenic protein oligomer A11) in the cortical (B) and hippocampal (C) region of indicated groups. (D) Quantification analysis of total Aβ and protein oligomer in the cortical and hippocampal region of indicated groups. Relative intensity of each band was determined by densitometry of dot blots (B and C) using ImageJ software and normalized to vehicle-administered APP/PS1 mouse group. All data are representative results of at least three independent experiments.

E Sandwich ELISA of Aβ-soluble and Aβ-insoluble fractions from cortical and hippocampal lysates of APP/PS1 mice. Three brain lysates per group were analysed.

F Western blot analyses of cortical and hippocampal lysates from APP/PS1 mice for APP and sAPPα expression.

G ThT assays for the inhibition of Aβ aggregation. Fluorescence intensity was normalized to Aβ aggregates (100%, day 5). Data are representative results of at least three independent experiments.

H SDS–PAGE analysis of PICUP cross-linked Aβ aggregates visualized by silver staining.

Data information: In (D, E and G), data are presented as mean ± SEM. *$P \leq 0.05$, **$P \leq 0.01$ and ***$P \leq 0.001$ (one-way ANOVA followed by Bonferroni's *post hoc* comparisons tests). Exact *P*-values are shown in Table EV4.

Source data are available online for this figure.

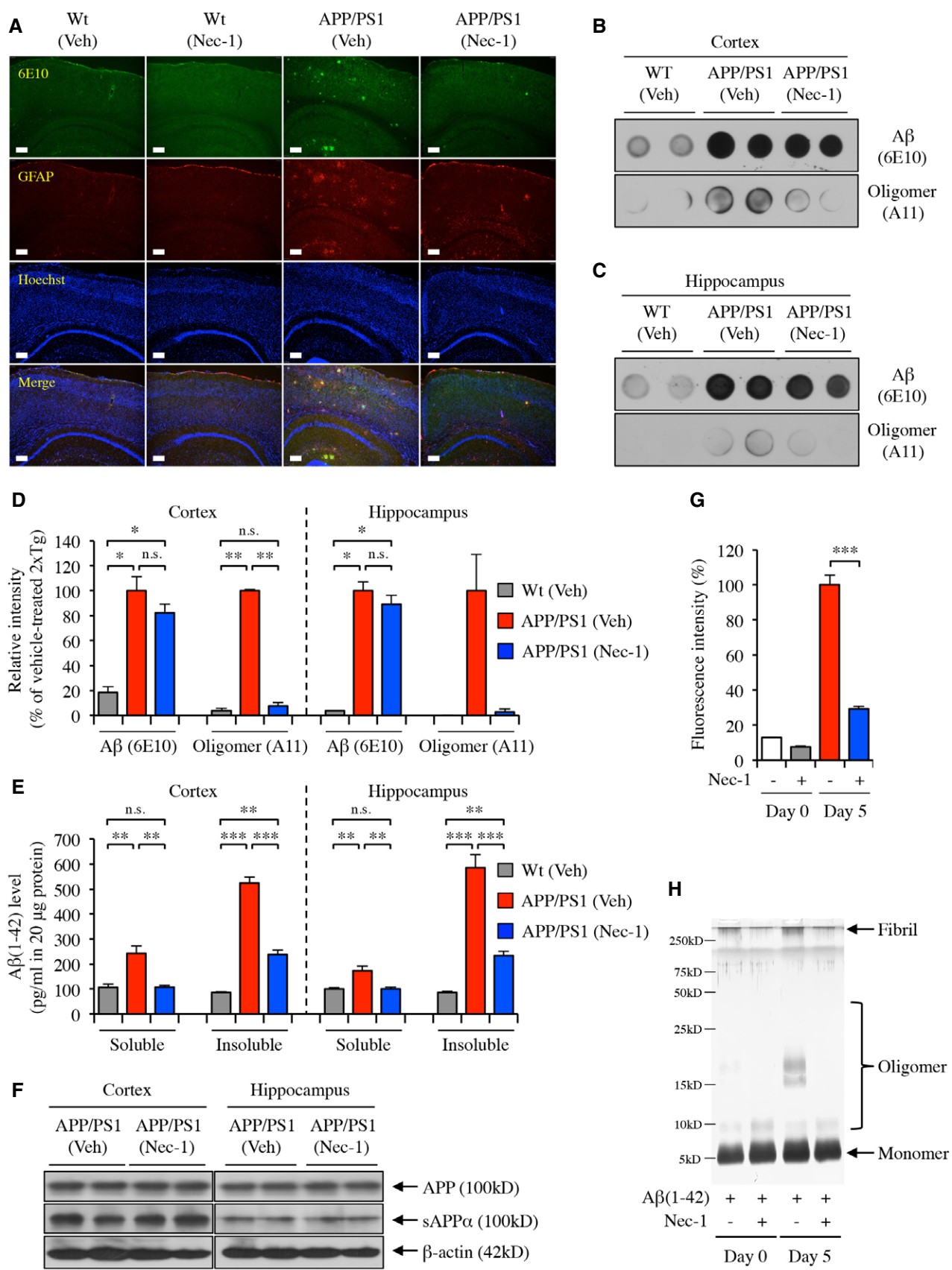

Figure 5.

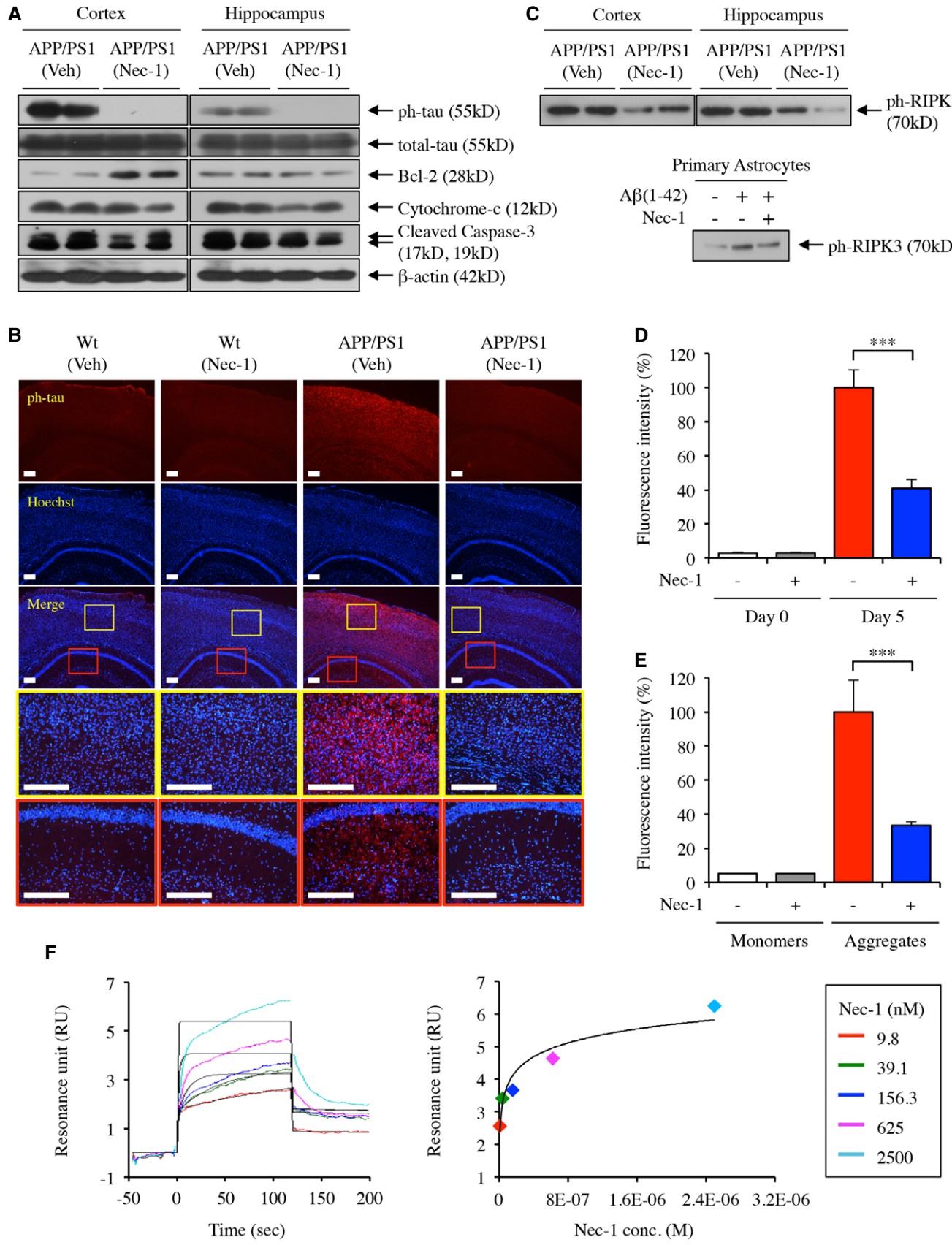

**Figure 6.**

**Figure 6. Nec-1 inhibits the phosphorylation and aggregation of tau.**

A    Western blot analysis of phosphorylation levels of tau and expression levels of indicated proteins in the cortical and hippocampal regions of the brain in APP/PS1 mice after administration of Nec-1 (6.25 mg/kg) or vehicle (2.5% DMSO in PBS). The membranes analysed are identical to those in Fig 5F.

B    Immunohistochemical analysis of cortical and hippocampal regions of the brains in wild-type (Wt) and APP/PS1 mice after administration of Nec-1 (6.25 mg/kg) or vehicle (2.5% DMSO in PBS) for phosphorylated tau expression levels. Anti-tau (phospho-Ser199) antibody (red colour) was applied for staining tau phosphorylation on serine-199 and Hoechst 33342 (blue colour) for nuclear counterstaining in the brain sections. Scale bars = 200 μm. Yellow and red boxes indicate the cortical and hippocampal region of brains, respectively.

C    Western blot analyses of phosphorylated RIPK3 expression in cortex and hippocampus of APP/PS1 mice injected with Nec-1 (upper), and primary cultured astrocytes treated with Nec-1 (lower). The membranes analysed are identical to those in Figs 1G and 5F.

D, E    ThT assays for the inhibition of tau aggregation (D) and for disaggregation of tau (E). Fluorescence intensity was normalized to tau aggregates (100%, day 5). All data are representative results of at least three independent experiments.

F    Surface plasmon resonance sensorgrams of Nec-1 targeting tau (left), and the corresponding dissociation fitting curve from the saturated region (right) under various concentrations of Nec-1.

Data information: In (D and E), data are presented as mean ± SEM. ***$P \leq 0.001$ (one-way ANOVA followed by Bonferroni's *post hoc* comparisons tests). Exact *P*-values are shown in Table EV5.

Source data are available online for this figure.

via an amine coupling. Nec-1 was injected into microfluidic channel as an analyte at a concentration ranging from 9.8 to 2,500 nM. Although it is unusual to assess bimolecular interactions between a small molecule and misfolded tau proteins, the resulting, but precise, binding constant ($K_D$ = 73.7 nM) indicated substantial direct interaction between Nec-1 and the K18 fragment (Fig 6F). Collectively, these findings suggested that Nec-1 blocks tau phosphorylation on Serine-199 and inhibits aggregate formation via direct interaction.

## Discussion

Here, we report that (i) Nec-1, a small molecule originally identified as an inhibitor of necroptosis, blocks Aβ-induced cell death in the brain; (ii) administration of Nec-1 ameliorates learning and memory deficits in the APP/PS1 mice; and (iii) Nec-1 reduces amyloid plaques and oligomers as well as tau hyperphosphorylation in the cortex and hippocampus of APP/PS1 mice.

Alzheimer's disease is characterized by four distinctive hallmarks, including Aβ plaques, neurofibrillary tangles, neural and synaptic loss, and cognitive impairments (Serrano-Pozo *et al*, 2011). With no overt symptoms of cognitive impairments until substantial levels of Aβ aggregation and tau hyperphosphorylation are accumulated in pre-symptomatic stage, however, AD is mostly diagnosed in later stages when irreversible degree of neural cell death and brain atrophy make full recovery from AD impractical (Jack *et al*, 2010, 2013). In addition, there are currently no fundamental therapies in symptomatic stage to effectively forestall the progression of AD (Selkoe, 2012). Thus, investigation of prophylactic approach to treat AD has been emphasized in many other studies, such as those of Anti-Amyloid Treatment in Asymptomatic Alzheimer's (A4) and Dominantly Inherited Alzheimer Network (DIAN), to prevent or delay the onset and progression of AD (Deweerdt, 2011; Garber, 2012; Selkoe, 2012).

In this study, we propose that Nec-1 regulates multiple pathological culprits that are important in AD and suggest drug repositioning to regulate all aspects of AD in a comprehensive manner, from underlying neuropathological changes to outward symptomatic outcomes. However, the detailed signalling mechanism by which Nec-1 regulates aforementioned hallmarks along with neural cell death remains to be determined. Many drugs have been developed to enhance AD cognitive deficits, such as memantine, which acts as

an NMDA receptor antagonist to block glutamate-mediated neurotoxicity (Danysz & Parsons, 2003). Given that memantine administration hardly halts AD progression, it is still debatable if the anti-cell death approach will be fundamentally effective in clinical treatments. Although not specifically examined in the AD animal model, previous study reporting that decreases in expression levels of murine Aβ and tau proteins by Nec-1 administration in aluminium-induced mouse model supports our view that Nec-1 has multiple mechanisms on AD pathology (Qinli *et al*, 2013). Further in-depth study on structure–activity relationships would facilitate interpretation of how Nec-1 can selectively target and regulate Aβ and tau aggregation. Moreover, prophylactic effects of Nec-1 may be limited to early stages of AD, based on the clinical failures of previous drug candidates inhibiting Aβ/tau abnormalities or brain cell death. Besides Nec-1 activity, we also observed the bimolecular binding between RIP kinase and Aβ. Interaction between these two proteins in the extracellular and luminal regions might provide Aβ-related cell death pathology in AD and also provide a potential drug target for the disease. Given that RIPK1/RIPK3 complex exhibits classical characteristics of amyloid aggregation, it is possible to predict that amyloidogenic properties of RIPK3 and Aβ attract each other. Additional studies are warranted to investigate whether the therapeutic effect of Nec-1 will translate into an intervention that might be potentially useful for AD.

## Materials and Methods

### Animals

APP/PS1 double-transgenic mice (male, APP/PS1, strain name; B6.Cg-Tg (APP$_{swe}$, PS1dE9) 85Dbo/J) and wild-type (male, C57BL/6) mice were obtained from the Jackson Laboratory (Bar Harbor, Maine, USA). In all experiments, the genotype of both APP and PS1 was confirmed via PCR analysis of tail DNA following the standard PCR condition from Jackson Laboratory. To generate Aβ (1–42) infusion mice model, 8-week-old male ICR mice were purchased from Orient Bio Inc. (Seoul, Korea) and habituated for 4 days. All mice were housed in a laboratory animal breeding room at the Korea Institute of Science and Technology and were maintained under controlled temperature with an alternating 12 h light–dark cycle and access to food and water *ad libitum*. At the

beginning of the experiment, 27 mice were assessed at an age of 4 months, APP/PS1 ($n = 13$, male) and wild type ($n = 14$, male). All animal experiments were performed in accordance with the National Institutes of Health guide for the care and use of laboratory animals (NIH Publications No. 8023, revised 1978). The animal studies were approved by the Institutional Animal Care and Use Committee of Korea Institute of Science and Technology.

## Reagents and antibodies

Aβ(1–42) peptides were synthesized by DMSO-incorporated Fmoc solid phase peptide synthesis (SPPS) protocol as described previously (Choi *et al*, 2012). Necrostatin-1 (Nec-1, Fig 1A), thioflavins S (ThS) and poly-D-lysine (PDL) were purchased from Sigma-Aldrich. Antibodies used for immunoblotting or co-immunoprecipitation were anti-amyloid-β (clone 6E10, Catalog SIG-39320, Covance), anti-amyloidogenic protein oligomer A11 (Catalog AHB0052, Invitrogen), anti-amyloid precursor protein A4 (clone 22C11, Catalog MAB348, Millipore Corporation), anti-sAPPα (clone 2B3, Catalog #11088, IBL), anti-PARP1 (Catalog sc-7150, Santa Cruz Biotechnology), anti-cleaved caspase-3 (Catalog #9661, Cell Signaling Technology), anti-cytochrome-c (Catalog sc-7159, Santa Cruz Biotechnology), anti-Bax (Catalog #2772, Cell Signaling Technology), anti-Bcl-2 (Catalog #2876, Cell Signaling Technology), anti-ph-JNK (Catalog #9251, Cell Signaling Technology), anti-JNK (Catalog #9252, Cell Signaling Technology), anti-ph-RIPK3 (Catalog ab209384, Abcam), anti-RIPK3 (Catalog IMG-5523, Imgenex), anti-tau (phospho-Ser199) (Catalog ab81268, Abcam), anti-tau (Catalog ab64193, Abcam) and anti-β-actin (Catalog MAB1501, Millipore Corporation).

## Cell culture

HT22 and BV2 cells were purchased from the Korean Cell Line Bank (Seoul National University, Republic of Korea) and cultured in culture medium (DMEM supplemented with 10% (vol/vol) foetal bovine serum (Gibco), 100 units/ml penicillin, 50 μg/ml streptomycin). Primary astrocytes were prepared from newborn C57BL/6 mice between P0 and P3 as previously described (Woo *et al*, 2012). The cerebral cortex and hippocampus were dissected and minced in culture medium (DMEM medium supplemented with 25 mM glucose, 10% (vol/vol) heat-inactivated horse serum, 10% (vol/vol) heat-inactivated foetal bovine serum, 2 mM glutamine and 1,000 units/ml penicillin–streptomycin). Then, single-cell suspension was prepared by pipetting gently. The astrocytes were then plated into 10-cm dish pre-coated with 0.1 mg/ml poly-D-lysine (PDL). On day 3 of culture, cells were vigorously washed with PBS and supplied with fresh culture medium. On the next day, cells were seeded for the experiments at a proper density of cells in fresh culture medium. All cultures were maintained at 37°C in a humidified 5% $CO_2$ incubator.

## Aβ(1–42) and Nec-1 treatment of cells

HT22, BV2 cells ($5 \times 10^5$ cells/well) and primary astrocytes ($2 \times 10^6$ cells/well) were plated into a six-well plate before treatment of Aβ(1–42) and Nec-1. Aβ(1–42) was dissolved in DMSO as 10 mM stocks. Stocks were then diluted 10-fold by cell starvation medium (0.5% foetal bovine serum in DMEM) and incubated for 24 h in 37°C to aggregate Aβ(1–42). Pre-aggregated Aβ(1–42) was applied to the cells for 12 and 24 h at concentration of 10 μM. Nec-1 (50 μM) was applied to the medium 15 min prior to application of Aβ(1–42) aggregates.

## Cell viability assays

Cultured cells were assessed for cytotoxicity of Aβ by MTT assay as previously reported (Kim *et al*, 2010). HT22, BV2 cells ($5 \times 10^3$ cells/well) and primary astrocytes ($2 \times 10^4$ cells/well) were seeded into a 96-well plate. After treatment of Aβ(1–42) aggregates and Nec-1, MTT reagent (15 μM) was added to each well and incubated for additional 4 h. 100 μl of solubilization solution was added and kept on room temperature for 16 h. The insoluble formazan was measured using EnSpire® Multimode Plate Reader (Perkin-Elmer) at 570 nm wavelength. Cell viability was also visualized and quantified by staining with the LIVE/DEAD® Viability/Cytotoxicity Assay kit (Molecular Probes) following the manufacturer's instructions.

## Immunoprecipitation and immunoblot analysis

Cells were lysed in ice-cold NP-40 lysis buffer (1% NP-40, 50 mM Tris–HCl, pH 7.5, 150 mM NaCl, 1 mM EDTA and protease inhibitor cocktail) for 15 min and centrifuged at 16,000 $g$, 4°C for 30 min. The supernatant of cell lysates were then pre-cleared with Sure-Beads Protein G Magnetic Beads (Bio-Rad) at 4°C for 4 h. After pre-clearance, the lysates were incubated with primary antibody at 4°C for 16 h. The antibody and lysates mixture were added with Sure-Beads Protein G Magnetic Beads at 4°C for 4 h. The immunoprecipitates were then washed with lysis buffer for six times and eluted by boiling in SDS sample buffer. For Western blot analysis, lysates (20 μg) were subjected to SDS–PAGE and transferred to a PVDF membrane. Membranes were treated with primary antibodies (1:1,000 dilution) overnight at 4°C and were detected by horseradish peroxidase-conjugated secondary antibodies. Blots were developed with the Clarity Western ECL Substrate (Bio-Rad) following the manufacturer's instructions.

## Docking simulation of Nec-1 and Aβ complex

### Preparation of ligands and receptor
The complex structure of RIPK1 with Nec-1 analog, Nec-1a (code: 4ITH) and the structure of 12-mer Aβ fibrils (code: 2LMO) were obtained from the PDB and then prepared using Protein Preparation Wizard of the Schrödinger Maestro program (Maestro, 2010). All water molecules were removed from the structure, and it was selected as a template. The structures of ligand Nec-1 were drawn on the basis of the conformation of Nec-1a in 4ITH, and their 3D conformations were generated using the Schrödinger LigPrep program with the OPLS 2005 force field (LigPrep, 2010).

### Molecular docking
Glide docking program in Schrödinger package was used to predict reasonable binding poses of Nec-1 within the hydrophobic site of RIPK1 and 12-mer Aβ fibrils. Docking performance was tested through redocking of the Nec-1 analog, Nec-1a, separated from the complex structure into the prepared RIPK1 (the score was −9.873), and the docking score of Nec-1 was similar (−9.550). RMSD values

(heavy atoms) were between 0.376 and 1.694 Å (average RMSD; 1.046 Å, median RMSD; 1.087 Å). Because the redocking test was successful, docking poses at other sites could be accepted as plausible bioactive conformations.

### In vivo experiments

#### APP/PS1 double-transgenic mouse model

Nec-1 (6.25 mg/kg) was injected into the tail vein twice a week as described in Fig 2A. In a Y-maze, spatial working memory was tested by recording spontaneous alternation behaviour after 12 weeks of Nec-1 injections. The apparatus was made of black plastic and composed of three equally spaced arms (40 L × 10 W × 12 H cm) labelled A, B and C that converged to the middle. Each mouse was placed at the end of one of the arms and was allowed to move freely for a 12-min session. An arm entry was defined as all four limbs of the mouse being within the arm completely. Entries into each arm were manually recorded for all mice. An alternation was defined as an entry different from the last two entries, and spontaneous alternation behaviour was calculated according to the following equation:

$$\% \text{ alternation} = 100 \times [(\text{number of alternations})/ \\ (\text{total number of arm entries} - 2)]$$

The passive avoidance test was performed 3 days after the Y-maze test. The apparatus consisted of a box divided into a bright compartment and a darker compartment with a grid floor. Each mouse was tested on two consecutive days. On the first day of testing, the training trial, each mouse was placed in the illuminated chamber and allowed to explore for 20 s. After 20 s, the wooden door between the light and dark compartments was opened. When the mouse entered the dark chamber with all four paws, the door closed and a foot shock (0.3 mA, 2 s, once) was delivered through the grid. The mouse was returned to its home cage. After 24 h, the second day of testing, each mouse was returned to the light chamber. When the mouse entered the dark chamber, the door was closed, and the step-through latency was measured with a cut-off time of 300 s. Data recordings of all behavioural responses were manually interpreted and converted into a computer-acceptable format by research colleagues in a blind experiment.

#### Aβ(1–42) infusion mouse model

Aβ(1–42) (100 μM) in PBS (10% dimethyl sulphoxide (DMSO)) was incubated at 37°C for 1 week to produce soluble Aβ aggregates (Kim et al, 2013a). Mice (n = 8 per each group, male) were subjected to the first intravenous injection of Nec-1 (6.25 mg/kg) 1 week before the acute intracerebroventricular injection of Aβ(1–42) aggregates (5 μl) or vehicle (10% DMSO in PBS). Nec-1 injections were given twice a week for 2 weeks.

### ThS staining and immunostaining

After behavioural tests, the brains of the wild-type and APP/PS1 mice were removed and fixed in 4% paraformaldehyde (pH 7.4). The fixed brain samples were then immersed in 30% sucrose for cryoprotection and cut into 30-μm-thick slices using a Cryostat (Microm HM 525, Thermo Scientific, Waltham, MA, USA). The sliced brains were stained with 500 μM of thioflavin S (ThS) dissolved in 50% ethanol for 7 min. The sections were then rinsed with 100, 95, 70% ethanol and PBS successively. For immunostaining, the slides were stained with anti-Aβ monoclonal antibody (1:100 dilution, clone 6E10, Catalog SIG-39320, Covance), anti-GFAP polyclonal antibody (1:300 dilution, Catalog AB5541, Millipore Corporation) and anti-tau (phospho-Ser199) antibody (1:100 dilution, Catalog ab81268, Abcam). Alexa Fluor 488- or Alexa Fluor 568-conjugated secondary antibodies (Life Technologies) were used for fluorescence detection. Hoechst 33342 (Sigma-Aldrich) was used for the visualization of nuclei. Images were taken on a Leica DM2500 fluorescence microscope (Li et al, 2007).

### Immunoblot analysis from brain lysates

After behavioural tests, the cortical and hippocampal regions were dissected separately from brains of wild-type and APP/PS1 mice. Brain tissues were then homogenized in RIPA buffer (20 mM Tris–HCl, pH 7.5, 50 mM NaCl, 0.5% NP-40, 4 mM EDTA, 0.1% SDS, 0.5% sodium deoxycholate and protease inhibitor cocktail) on ice for 15 min and centrifuged at 16,000 g, at 4°C for 30 min. The supernatant of brain lysates was then used for Western blot and dot blot analysis. The concentrations of lysates were quantified by BCA assay. For Western blot analysis, the brain lysates (20 μg) were subjected to SDS–PAGE and transferred to a PVDF membrane. 20 μg of brain lysates were spotted for the dot blot analysis to detect amyloidogenic protein oligomer and Aβ. Membranes were treated with primary antibodies (1:1,000 dilution) overnight at 4°C and were detected by horseradish peroxidase-conjugated secondary antibodies. Blots were developed with the Clarity Western ECL Substrate (Bio-Rad) following the manufacturer's instructions.

### Quantification of soluble and insoluble Aβ from brain lysates

After acquiring brain lysates by using RIPA buffer (soluble fraction), the pellet of brain lysates was then lysed in guanidine buffer (5 mM guanidine-HCl, 50 mM Tris–HCl pH 8.0) containing proteinase inhibitor cocktail. After the incubation at room temperature for 3 h on a shaker to dissolve Aβ-insoluble fraction, the mixtures were centrifuged at 4°C for 2 h to obtain insoluble fraction from the supernatant. Levels of Aβ(1–42) dissolved in soluble and insoluble fractions were measured by Aβ(1–42) sandwich ELISA kit purchased from Invitrogen (KHB3442). 20 μg of protein samples was used in sandwich ELISA, which was performed according to the manufacturer's instructions.

### Inhibition of Aβ aggregation

In-house synthetic Aβ(1–42) peptides (25 mM) were dissolved in DMSO and diluted with deionized water to make Aβ(1–42) solutions (25 μM). Nec-1 (500 μM) was added before the solutions were incubated for 5 days at 37°C. At the end of incubation period for aggregation, Aβ oligomerization was monitored using the thioflavin-T (ThT) assay. ThT (5 μM in 50 mM glycine buffer, pH 8.9) was added in 96-well black plate and incubated for 3 h. Fluorescence of

Aβ-bound ThT was measured at 450 nm (excitation) and 485 nm (emission) using EnSpire® Multimode Plate Reader (Perkin-Elmer).

### SDS–PAGE with photo-induced cross-linking of the unmodified proteins (PICUP)

To analyse Aβ species by size distribution, we performed SDS–PAGE and PICUP chemistry (Bitan & Teplow, 2004). Aβ solutions were quickly irradiated for 1 s, twice, in order to cross-link Aβ peptides with Ru(Bpy)(Cl2) and ammonium persulphate. Cross-linked Aβ samples were examined on 15% tris-tricine gels. After separation, peptide bands on gels were visualized using silver-staining kit.

### Inhibition of tau aggregation/tau disaggregation assays

The K18 fragment (125 amino acids, 0.5 mg/ml), which was cloned from full-length human tau, was used for induction of tau aggregation as previously reported (Haque *et al*, 2014). The tau K18 fragment (35 μM) in phosphate-buffered saline (PBS, pH 7.4) was incubated with 0.1 mg/ml heparin (Sigma) and 100 μM dithiothreitol (DTT) (Sigma) at 37°C for 5 days. Nec-1 (50 μM) was added to the aggregation mixtures prior to a 5-day incubation period (for the inhibition assay of tau aggregation) or after a 5-day incubation period (for the tau disaggregation assay). At the end of incubation period for aggregation, ThT (5 μM in 50 mM glycine buffer, pH 8.9) was added in 96-well black plate and incubated for 3 h. Fluorescence of tau-bound ThT was measured at 450 nm (excitation) and 485 nm (emission) using EnSpire® Multimode Plate Reader (Perkin-Elmer).

### Surface plasmon resonance analysis

The surface plasmon resonance analysis was performed using Biacore T200 instrument and Series S carboxymethylated (CM5) dextran matrix sensor chips (GE Healthcare) (Richter *et al*, 2010). HBS-EP$^+$ (10 mM HEPES, pH 7.4, 150 mM NaCl, 3 mM EDTA and 0.05% surfactant P20) was used as a running buffer at 25°C. Aβ(1–42) aggregates (200 μg/ml) and tau (1 mg/ml) were diluted with 10 mM sodium acetate solution (pH 4.0 and pH 5.5, respectively) to make 40 μg/ml of Aβ(1–42) aggregates and 25 μg/ml of tau. Then, they were covalently immobilized on the chip surface by amine coupling chemistry. The remaining activated carboxymethyl groups on the surface were blocked by injection of 1 M ethanolamine (pH 8.0). Immobilization values of Aβ(1–42) aggregates and tau were 6,000 RU and 1,400 RU, and theoretical Rmax of Aβ(1–42) aggregates and tau were 386 RU and 29 RU, respectively. Nec-1 was prepared in PBS-T running buffer (10 mM phosphate, 135 mM NaCl, 27 mM KCl and 0.005% surfactant P20) containing 1% DMSO as serial-diluted samples. For the binding and kinetics assay, Nec-1 was injected for 120 s at a flow rate of 30 μl/min.

### Statistical analysis

Graphs were obtained with GraphPad Prism 6, and the statistical analyses were performed with one-way ANOVA followed by Bonferroni's *post hoc* comparisons ($*P < 0.05$, $**P < 0.01$, $***P < 0.001$). The error bars represent the SEM.

**The paper explained**

**Problem**

Alzheimer's disease (AD) is a progressive neurodegenerative disease with multiple putative therapeutic targets. Abnormal Aβ and Tau species are considered to be strong candidates for the development and progression of AD, but by which exact mechanisms remain unclear. Thus, most of the currently available drugs are aimed at providing more practical benefits to mask the cognitive symptoms of AD in patients. However, without affecting the underlying causes of AD, the disease will not be delayed or ultimately treated. It is also challenging to develop AD therapeutics aimed at the symptomatic stage given that Aβ and Tau abnormalities begin in pre-symptomatic stage, followed by irreversible neural cell death, but before outward cognitive impairments are observed in patients. Means to modulate brain atrophy prophylactically could be an interesting strategy against AD.

**Results**

In this study, we report that Nec-1, which was originally identified as a necroptosis inhibitor, is capable of targeting multiple hallmarks of AD from pathophysiological changes to behavioural outcomes. Nec-1 is found to directly interact with Aβ aggregates and prevent Aβ-induced neural cell death *in vitro* when different types of neural cells were pre-treated with Nec-1 before the addition of Aβ aggregates in the culture media. The study also provides *in vivo* evidence that i.v. injection of Nec-1 before the onset of AD-like phenotypes significantly reduces Aβ oligomers, plaques and hyperphosphorylated tau in the cortex and hippocampus, alters apoptotic marker protein expression levels and inhibits cognitive impairments in AD mouse models.

**Impact**

Nec-1 can modulate multiple culprits of AD, from alleviating progressive cognitive impairments (through preventing neurodegeneration) to treating amyloidal properties of Aβ and Tau in the AD brain.

**Expanded View** for this article is available online.

## Acknowledgements

This research was supported by National Research Council of Science & Technology (NST, CRC-15-04-KIST), Basic Science Research Program through the National Research Foundation of Korea (NRF, 2015R1A6A3A04058568 and 2014R1A1A3051648) funded by the Ministry of Education, Science and Technology, and Korea Institute of Science and Technology (KIST Young Fellowship, 2V05030). The authors thank Mr. Yakdol Cho (Korea Institute of Science and Technology) for animal maintenance and preparation; Dr. Yun Kyung Kim (Korea Institute of Science and Technology) and Dr. Sungsu Lim (Korea Institute of Science and Technology) for preparation of tau aggregation; Mr. Juhyeong Jo (GE Healthcare Korea/Japan) for Biacore analyses; and Ms. Sarah Hesse (University of Glasgow) for editing advices. The authors appreciate Dr. Hye Yun Kim for scientific advices.

## Author contributions

S-HY and YK designed the experiments. S-HY made a major contribution in all the experiments. JS and JK performed animal behavioural test and surface plasmon resonance analysis. SL and SB prepared synthetic Aβ(1–42). HJ and J-MH performed structural modelling analysis. S-HY, DKL and YK wrote the manuscript.

## Conflict of interest

The authors declare that they have no conflict of interest.

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
