## [Review Process File · EMBO Molecular Medicine]

Nec-1 alleviates cognitive impairment with reduction of A and tau abnormalities in APP/PS1 mice

Seung-Hoon Yang, Dongkeun Kenneth Lee, Jisu Shin, Sejin Lee, Seungyeop Baek, Jiyeon Kim, Hoyong Jung, Jung-Mi Hah, and YoungSoo Kim

Corresponding author: YoungSoo Kim, Korea Institute of Science and Technology

Review timeline:

Submission date:	22 April 2016
Editorial Decision:	16 June 2016
Revision received:	07 September 2016
Editorial Decision:	23 September 2016
Revision received:	11 October 2016
Accepted:	17 October 2016

Transaction Report:

Editor: Céline Carret

1st Editorial Decision

16 June 2016

Thank you for the submission of your manuscript to EMBO Molecular Medicine. We have now heard back from the three referees who were asked to evaluate your manuscript. Although the referees find the study to be of potential interest, they also raise a number of concerns that need to be addressed in the next final version of your article.

You will see from their comments below that all three referees find the data intriguing. Unfortunately, lack of appropriate controls in many places, as well as limited mechanistic details reduce the significance of the work.

Given these evaluations, I would like to give you the opportunity to revise your manuscript, with the understanding that the referee concerns must be fully addressed and that acceptance of the manuscript would entail a second round of review. We would like to particularly encourage you to add all controls and appropriate methodology/antibody and provide some experimental evidence of the mode of action of Nec-1 in this setting.

Please note that it is EMBO Molecular Medicine policy to allow only a single round of revision and that, as acceptance or rejection of the manuscript will depend on another round of review, your responses should be as complete as possible. Revised manuscripts should be submitted within three months of a request for revision; they will otherwise be treated as new submissions, except under exceptional circumstances in which a short extension is obtained from the editor. Also, please consult our guidelines and format your revised article accordingly.

I look forward to seeing a revised form of your manuscript as soon as possible.

***** Reviewer's comments *****

Referee #1 (Remarks):

The current study by Yang et al reports that necrostatin-1 (Nec-1), an anti-necrotic molecule, regulates four different pathological and behavioral hallmarks of Alzheimer's disease (AD) in 2xTg-AD mice. With increasing interest and potential benefits from a single molecule with therapeutic effects in AD, results of this study will provide information useful to the field, especially in the development of AD drugs. It is significant findings for the fact that Nec-1 is capable of targeting both A β and tau proteins to further affect cognitive ability of AD mouse model, and it is also intriguing that authors described the direct binding relationships between Nec-1 with these proteins. However, there are several concerns of which will improve quality of their study once they are answered before publication. Detailed concerns are as follows:

1. In Figure 6B-C, authors showed that Nec-1 reduces amount of hyperphosphorylated tau using an artificially altered cell line. My concerns are that these are not sufficient, and moreover such cell line is rather inappropriate to strongly support the results. As authors visually demonstrated the reduction of A β aggregates using cell tissues, I highly recommend authors to confirm the inhibition of tau phosphorylation using results from animal models.
2. Nec-1 toxicity was not studied experimentally. For Nec-1 to be considered as a potential drug candidate, preliminary data showing non-cytotoxicity of Nec-1 needs to be included. Please provide evidence that Nec-1 functions with no apparent cytotoxicity in both cell and animal models.
3. As mentioned in the introduction, Nec-1 regulates necroptosis by serving as a RIP kinase inhibitor. Authors only showed that apoptotic neural cell deaths were inhibited by Nec-1, with no supporting results in regards to its function in necroptosis. Please demonstrate that Nec-1 indeed inhibits the RIP kinases and provide an explanation for why authors focused on effect of Nec-1 on apoptotic neural cell deaths.

Minor comments:

1. Material and Methods: please provide detailed information about antibodies.
2. Immunostaining and immunoblotting Protocols: provide more details including concentrations of antibodies and incubation times.
3. Figure 1A: cite the reference if it is not drawn by the author.

Referee #2 (Comments on Novelty/Model System):

The study is mainly suffering from the lack of controls.

Referee #2 (Remarks):

Though the hypothesis that NEC-1 might have a role in AD might be interesting, the lack of controls decreases the impact of any finding reported. The authors claim that cell death was induced by Abeta42 aggregates that were produced by incubating Abeta peptides for 24hrs. Such material was not characterized before used in the study but usually after 24hrs Abeta42 has formed fibrils that are not toxic (controls are lacking; e.g., scrambled Abeta). It is unclear how NEC-1 treatment was done (control lacking).

Bimolecular interaction: NEC-1 and Abeta or Tau might unspecifically interact at such high concentrations used (10 μ M). This view is supported by SPR curves shown in figures 3A and 6F. All curves show an unusual shape that indicate an unspecific aggregation of material on the chip surface rather than a binding reaction (controls are lacking).

Animal treatment: controls are lacking

Referee #3 (Remarks):

The manuscript titled, "Nec-1 alleviates cognitive impairment with reduction of A β and tau abnormalities in APP/PS1 mice" by Tang et al., describes the therapeutic effects of necrostatin-1 (Nec-1), necroptosis inhibitor against the cognition and pathology of Alzheimer's disease.

The manner of neuronal cell death in the brain of AD patients is still debatable. The authors administered Nec-1, necroptosis inhibitor, to APP/PS1 mice and found that administration of Nec-1 ameliorated memory deficit and reduced A β deposition and the level of phospho-tau. The authors also found that Nec-1 reduced cellular toxicity of A β aggregates, Nec-1 inhibited the phosphorylation in culture cells. Those data suggested the Nec-1 might have a potential clinical benefit in Alzheimer disease. On the other hand, this work is too immature and the authors did not address several important points to understand the mechanism whereby Nec-1 reduced A β deposition and phosphorylation of tau.

1. The authors should show the evidence whether Nec-1 entered in the brain of APP/PS1 mice and affect against neurons. Nec-1 is known as an inhibitor of necroptosis, thus, the authors may quantify the level of autophosphorylation of RIP1 in the brain of APP/PS1 mice.
2. To check whether Nec-1 affect the production of A β , the authors should check the level of APP and sAPP β in the brain of APP/PS1 mice. Also, the dot-blot data probed by 6E10 in Fig.5 is not clear. Although 6E10 recognizes human A β , not mouse A β or APP, the dot-blot exhibited signals from the brains of vehicle-treated wild-type mice. The authors should measure the level of soluble and insoluble A β using specific ELISA.
3. The author described the possibility that Nec-1 directly A β aggregates and prevents the formation of A β plaques at page 11. The possibility is very important. The authors should examine the effect of Nec-1 against the fibrillization of synthetic A β 1–42 by in vitro A β fibrillization assay using a thioflavin T.
4. In Fig.3A, the authors exhibited the data A β interacts with RIPK3. I wonder why extracellular/luminal A β is able to bound to cytoplasmic RIPK3. Please explain it more.
5. In Fig.6, the authors exhibited a nice data that Nec-1 reduced the level of phosphor-tau in the brain of APP/PS1 mice. Those mice possess the murine tau, not human tau. Thus, the author may examine the effects of Nec-1 against phosphorylation of tau and aggregation of tau using murine tau, not human tau.
6. 2xTg-AD is not usual to use. Please use APP/PS1 or APP^{swe}/PSEN1^{dE9} in the whole manuscript and refer the Dr. Jankowski's original paper.
7. It is well known there are gender differences in the amount and deposition of A β in the brain of APP tg mice. Please describe the gender of APP tg mice used in this work in the Materials and Methods.

1st Revision - authors' response

07 September 2016

Referee #1 (Remarks):

First of all, we would like to express our sincere gratitude for your thorough comments, which further enriched this paper. All authors reviewed the comments in details and responded with care. Thank you very much for your time and consideration.

The current study by Yang et al reports that necrostatin-1 (Nec-1), an anti-necrotic molecule, regulates four different pathological and behavioral hallmarks of Alzheimer's disease (AD) in 2xTg-AD mice. With increasing interest and potential benefits from a single molecule with therapeutic effects in AD, results of this study will provide information useful to the field, especially in the development of AD drugs. It is significant findings for the fact that Nec-1 is capable of targeting both A β and tau proteins to further affect cognitive ability of AD mouse model, and it is also intriguing that authors described the direct binding relationships between Nec-1 with these proteins. However, there are several concerns of which will improve quality of their study once they are answered before publication. Detailed concerns are as follows:

1. In Figure 6B-C, authors showed that Nec-1 reduces amount of hyperphosphorylated tau using an artificially altered cell line. My concerns are that these are not sufficient, and moreover such cell line is rather inappropriate to strongly support the results. As authors visually demonstrated the reduction of A β aggregates using cell tissues, I highly recommend authors to confirm the inhibition of tau phosphorylation using results from animal models.

→ Thank you for the comment. We understand that our data was not sufficient to support that Nec-1 reduces the amount of hyperphosphorylated tau. We conducted immunohistochemical analysis using APP/PS1 mice model to show the effect of Nec-1 on the inhibition of tau phosphorylation directly instead of using engineered cell line. Consistent with the result of western blot analysis, we confirmed that Nec-1 dramatically reduced the level of phosphorylated tau. We replaced this result as Figure 6B instead of former figure from HEK293-human tau-BiFC engineered cell line.

2. Nec-1 toxicity was not studied experimentally. For Nec-1 to be considered as a potential drug candidate, preliminary data showing non-cytotoxicity of Nec-1 needs to be included. Please provide evidence that Nec-1 functions with no apparent cytotoxicity in both cell and animal models.

→ We appreciate that you have pointed out what could help us make our manuscript more refined and sound. We examined the toxicity of Nec-1 on both cells and animal models. We found out that Nec-1 has no cytotoxic effect on the HT22 cell line according to MTT assays and Nec-1 administration does not have an effect on the survival rate of APP/PS1 mice model. These results made our manuscript more comprehensive in the sense that we have studied the possibility of Nec-1 as a drug candidate of AD, and they were added as Figure 1B and 2E.

3. As mentioned in the introduction, Nec-1 regulates necroptosis by serving as a RIP kinase inhibitor. Authors only showed that apoptotic neural cell deaths were inhibited by Nec-1, with no supporting results in regards to its function in necroptosis. Please demonstrate that Nec-1 indeed inhibits the RIP kinases and provide an explanation for why authors focused on effect of Nec-1 on apoptotic neural cell deaths.

→ Thank you very much for the critical comment. Nec-1 is well-known for its function in relation to necroptosis, so it is necessary to study whether Nec-1 inhibits the RIP kinases in neural cells. We apologize for having no evidence included in our previous manuscript. We examined the reduction in the phosphorylation of RIPK3, which is a member of RIP kinase family, by Nec-1. The results showed that Nec-1 significantly reduces the levels of phosphorylated RIPK3 in both cortex and hippocampus of APP/PS1 mouse brains as well as primary cultured astrocytes. These results were provided as Figure 6C. We also tried to examine the phosphorylation of RIPK1; however, an antibody against murine phosphorylated RIPK1 was not currently available. Alternatively, we conducted two different methods; 1) we performed co-immunoprecipitation using an antibody against RIPK1 followed by phosphor-serine antibody from brain lysates of APP/PS1 mice model, and 2) we used an antibody against human phosphorylated RIPK1 to expect cross-reactivity with murine phosphorylated RIPK1. Unfortunately, phosphorylated RIPK1 was not detectable by both methods. Although it was not possible to directly show Nec-1 affecting the phosphorylation of RIPK1, we believe that our result of reduced level of phosphorylated RIPK3 is sufficient to provide evidence that Nec-1 serves its anti-necroptotic role in the brain of APP/PS1 mice.

Minor comments:

1. Material and Methods: please provide detailed information about antibodies.

→ We appreciate for the detailed comment. We provided more information about antibodies to include their catalog numbers and company names in “Reagents and Antibodies” part of Material and Methods.

2. Immunostaining and immunoblotting Protocols: provide more details including concentrations of antibodies and incubation times.

→ We apologize for lacking details on concentrations and incubations times for antibodies. We provided dilution factors for each antibody and incubation times (overnight at 4°C) in Material and Methods.

3. Figure 1A: cite the reference if it is not drawn by the author.

→ Thank you for the comment. Because Figure 1A was drawn by one of authors, we stated it as, “Chemical structure of Nec-1. The schematic diagram was drawn by S. Lee using ChemDraw Professional 15.0. software” in the legend of Figure 1A.

Referee #2 (Comments on Novelty/Model System):

First of all, we would like to express our sincere gratitude for the time and effort the referee had put into reviewing our manuscript. All authors carefully went over the all the comments in details in order to further refine the paper. Thank you very much for your thorough considerations.

The study is mainly suffering from the lack of controls.

→ We apologize for insufficient information regarding controls in our study. We carefully followed concerns raised and responded by adding and correcting controls in experiments included this study in point-by-point fashion. The corrections are listed as below. We also appreciate that the editor, Dr. Celine Carret, for informing us in the follow-up emails regarding details on what controls we could add in the manuscript.

Referee #2 (Remarks):

Though the hypothesis that NEC-1 might have a role in AD might be interesting, the lack of controls decreases the impact of any finding reported. The authors claim that cell death was induced by Abeta42 aggregates that were produced by incubating Abeta peptides for 24hrs. Such material was not characterized before used in the study but usually after 24hrs Abeta42 has formed fibrils that are not toxic (controls are lacking; e.g., scrambled Abeta).

→ Thank you for in-depth comments. We confirmed that A β synthesized in our laboratory contains more oligomers when incubated for 24 hours, by photo-induced cross-linking of unmodified proteins (PICUP) method as the figure provided below. This figure was not included in the manuscript.

[Data removed upon authors' request]

A β fibrils are known to be not toxic. However, this is more relevant when commercially purchased A β is incubated for 24 hours. For our study, we incubated A β that we had synthesized in our laboratory. When our own synthesized A β is incubated, not only fibrils but also toxic oligomers are produced, which is a result that we have already reported in the previous study [1-4]. Based on our previous studies, we are confident that our synthetic A β (1-42) is highly active in toxicity and amyloidogenic property and that our protocol regulating aggregation of A β in this study provided toxic oligomers to cell lines.

It is unclear how NEC-1 treatment was done (control lacking).

We apologize for providing insufficient information that it was confusing how Nec-1 treatment was processed. We corrected this issue by stating, “Nec-1 (50 μ M) was applied to each cell model (pre-treatment) 15 minutes before A β (1-42) aggregates (10 μ M) was added to cells” in Results (Line 2 of Page 5).

Bimolecular interaction: NEC-1 and Abeta or Tau might unspecifically interact at such high concentrations used (10 μ M). This view is supported by SPR curves shown in figures 3A and 6F. All curves show an unusual shape that indicate an unspecific aggregation of material on the chip surface rather than a binding reaction (controls are lacking).

We appreciate for your careful review. We agree that there are possibilities that high concentrations of A β and tau may lead to unspecific bindings to Nec-1. We confirmed that it was indeed the specific binding to Nec-1 by using a negative control. We utilized BSA, which is a commonly used negative control in SPR analysis [5], in high concentrations as same as A β and tau aggregates. Our result shown below did not show a dose-dependent curve, suggesting that unspecific binding was not present. This result was not included in the manuscript.

[Data removed upon authors' request]

Animal treatment: controls are lacking.

The authors have used one model of tgs with several FAD mutations combined. Since it is known that in any other genetically modified mouse model ~100 other genes are affected, the drug might cause effects in the specific model that is not seen elsewhere.

→ We deeply appreciate for the in-depth comment regarding animal treatment. In the study of a drug candidate, it is critical that the effect we observe is not specific to a certain genetically modified mouse model. In order to confirm that the effect of Nec-1 was not specific to only APP/PS1 mice model or genetically engineered rodents, we conducted animal study using wild-type ICR mice model that we have acutely infused A β (1-42) by intracerebroventricular (i.c.v.) injections without any genetic modifications [6]. Consistent with the results from APP/PS1 mice model, Nec-1 administration in A β (1-42) infusion mice improved cognitive deficits in behavioural Y-maze tests. We provided results as Figure 2F-I.

Referee #3 (Remarks):

First of all, we truly appreciate and express our sincere gratitude for the in-depth review and comments, which all the authors have responded with care to further enrich this paper. Our responses to your concerns have helped us to answer many potential concerns that could have been raised by other readers. Thank you very much for your time and consideration.

The manuscript titled, "Nec-1 alleviates cognitive impairment with reduction of A β and tau abnormalities in APP/PS1 mice" by Yang et al., describes the therapeutic effects of necrostatin-1 (Nec-1), necroptosis inhibitor against the cognition and pathology of Alzheimer's disease.

The manner of neuronal cell death in the brain of AD patients is still debatable. The authors administrated Nec-1, necroptosis inhibitor, to APP/PS1 mice and found that administration of Nec-1 ameliorated memory deficit and reduced A β deposition and the level of phospho-tau. The authors also found that Nec-1 reduced cellular toxicity of A β aggregates, Nec-1 inhibited the phosphorylation in culture cells. Those data suggested the Nec-1 may have a potential clinical benefit in Alzheimer disease. On the other hand, this work is too immature and the authors did not address several important points to understand the mechanism whereby Nec-1 reduced A β deposition and phosphorylation of tau.

1. The authors should show the evidence whether Nec-1 entered in the brain of APP/PS1 mice and affect against neurons. Nec-1 is known as an inhibitor of necroptosis, thus, the authors may quantify the level of autophosphorylation of RIP1 in the brain of APP/PS1 mice.

→ Thank you very much for bringing up a critical point. We apologize for providing insufficient information regarding whether Nec-1 has its effect in the brain of APP/PS1 mice. Blood-Brain-Barrier permeability of Nec-1 has been reported previously [7-9] and thus Nec-1 was used in several mice models to study brain disorders. We examined the effect of Nec-1 on the level of RIP kinase phosphorylation to deal with this issue. More specifically, we examined the effect of Nec-1 on the phosphorylation of RIPK3, which is a member of RIP kinase family, in the brain of APP/PS1 mice model. The results showed that Nec-1 significantly reduces the levels of phosphorylated RIPK3 in both cortex and hippocampus of APP/PS1 mouse brains as well as primary cultured astrocytes, which we added in Figure 6C. We also tried to measure RIPK1 phosphorylation; however, an antibody against murine phosphorylated RIPK1 was not currently available. Alternatively, we conducted two kinds of methods; 1) we performed co-immunoprecipitation using an antibody against RIPK1 followed by phosphor-serine antibody from brain lysates of APP/PS1 mice model, and 2) we used an antibody against human phosphorylated RIPK1 to expect cross-reactivity with murine phosphorylated RIPK1. Unfortunately, phosphorylated RIPK1 was not detectable by both methods. Although it was not possible to directly show that Nec-1 affects the phosphorylation of RIPK1, we believe that our result of reduced level of phosphorylated RIPK3 by Nec-1 is sufficient to provide evidence that Nec-1 affects against neural cells and serves its anti-necroptotic role in the brain of APP/PS1 mice.

2. To check whether Nec-1 affect the production of A β , the authors should check the level of APP and sAPP in the brain of APP/PS1 mice.

→ Thank you for pointing out an issue we may have missed without the thorough review. Corrections made in consideration of the comment would help us to refine our manuscript in a great

extent. We measured APP and sAPP α from brain lysates of APP/PS1 mice model using specific antibody against APP (22C11 clone, Catalog MAB348, Millipore Corporation) and sAPP α (2B3 clone, Catalog #11088, IBL). Results showed that Nec-1 did not lead to changes in APP and sAPP α expressions, suggesting that Nec-1 does not affect the production of A β . These results are added as Figure 5F.

Also, the dot-blot data probed by 6E10 in Fig.5 is not clear. Although 6E10 recognizes human A β , not mouse A β or APP, the dot-blot exhibited signals from the brains of vehicle-treated wild-type mice. The authors should measure the level of soluble and insoluble A β using specific ELISA.

→ We appreciate for your comment. We separately measured the levels of soluble and insoluble A β from brain lysates of APP/PS1 mice model by specific ELISA (KHB3442, Invitrogen). Consistent with our dot blot analysis, both soluble and insoluble A β levels were significantly reduced by Nec-1 in both cortex and hippocampus, though endogenous levels of A β were still observed from brain lysates of vehicle-treated wild-type mice. These results were added as Figure 5E.

3. The author described the possibility that Nec-1 directly A β aggregates and prevents the formation of A β plaques at page 11. The possibility is very important. The authors should examine the effect of Nec-1 against the fibrillization of synthetic A β by in vitro A β fibrillization assay using a thioflavin T.

→ Thank you for the comment. As stated in the comment, the possibility that Nec-1 directly binds to A β aggregates and prevent A β plaque formation is very important issue in our manuscript. We conducted ThT assay to examine whether Nec-1 indeed inhibits the fibrillization of synthetic A β *in vitro*. Results showed that fluorescence intensity was dramatically decreased when Nec-1 was incubated together with A β , suggesting that Nec-1 does inhibit A β aggregation. We further confirmed the inhibitory effect of Nec-1 on A β aggregation by PICUP analysis, which showed reductions in both A β fibrils and oligomers by Nec-1. The results were added as Figure 5G and H.

4. In Fig.3A, the authors exhibited the data A β interacts with RIPK3. I wonder why extracellular/luminal A β is able to bound to cytoplasmic RIPK3. Please explain it more.

→ The origin of our study is based on the structural and functional interpretation of protein complex formation between RIPK1 and RIPK3 [10]. According to Li *et al.*, the RIPK1/RIPK3 complex resembles the amyloid characteristics. In the discussion, we suggested that there is a possibility of direct bimolecular interaction between A β and RIPK. We added in the Discussion as follows: “Besides Nec-1 activity, we also observed the bimolecular binding between RIP kinase and A β . Interaction between these two proteins in the extracellular and luminal regions might provide A β -related cell death pathology in AD and also provide a potential drug target for the disease. Given that RIPK1/RIPK3 complex exhibits classical characteristics of amyloid aggregation, it is possible to predict that amyloidogenic properties of RIPK3 and A β attract each other.”

5. In Fig.6, the authors exhibited a nice data that Nec-1 reduced the level of phosphor-tau in the brain of APP/PS1 mice. Those mice possess the murine tau, not human tau. Thus, the author may examine the effects of Nec-1 against phosphorylation of tau and aggregation of tau using murine tau, not human tau.

→ We appreciate for the careful review. We agree that we should have shown the effect of Nec-1 against murine tau phosphorylation instead of human tau. We further carried out immunohistochemical analysis of APP/PS1 mice model using an antibody against murine phosphorylated tau (Catalog ab81268, Abcam) to visualize the effect of Nec-1 on the inhibition of tau phosphorylation in the brain directly. Consistent with the result of western blot analysis, we confirmed that Nec-1 dramatically reduced the level of phosphorylated tau. We replaced this result as Figure 6B instead of the former figure from HEK293-human tau-BiFC cell line.

We understand why the referee commented that our study is short on the understanding of Nec-1 mechanism against Alzheimer and why the referee suggested to extend our work by utilizing murine tau. However, our goal is to develop a disease-modifying drug candidate to treat multiple pathogens of Alzheimer's disease. In this study, we showed that Nec-1 regulates aggregation and phosphorylation of human tau by ThT and cell-based assays, respectively. We also observed the direct binding between Nec-1 and human tau by SPR analyses. Fortunately, the human tau and murine tau share phosphorylation sites, which can be detected by the same antibodies such as anti-tau (phospho Ser199) antibody (Catalog ab81268, Abcam). Therefore, our findings, from the APP/PS1 mouse brains, support the view that Nec-1 directly binds to human tau and inhibits

phosphorylation of Serine-199, which is an important post-translational modification site related to tauopathy.

We regret that the murine tau does not form tangles like human tau and, therefore, we could not confirm inhibitory actions of Nec-1 against tau aggregation *in vivo*. However, we confirmed that Nec-1 inhibits phosphorylation of Serine-199 by immunohistochemical analyses. To avoid confusion, we toned down the interpretation on our findings from inhibition of tau phosphorylation to inhibition of Serine-199 phosphorylation in tau proteins.

6. 2xTg-AD is not usual to use. Please use APP/PS1 or APP^{swe}/PSEN1^{dE9} in the whole manuscript and refer the Dr. Jankowski's original paper.

→ We apologize for using terms that were confusing. We have corrected the term from “2xTg-AD” to “APP/PS1” throughout the manuscript and have referred to Dr. Jankowski's original paper (Line 12 of Page 7).

7. It is well known there are gender differences in the amount and deposition of A β in the brain of APP tg mice. Please describe the gender of APP tg mice used in this work in the Materials and Methods.

→ Thank you for the comment. Mice we used for this study were all male, and we have clarified their genders in the Results, Materials and Methods, and Figure legends.

References

1. Choi, J.W., et al., *Efficient access to highly pure beta-amyloid peptide by optimized solid-phase synthesis*. Amyloid-Journal of Protein Folding Disorders, 2012. **19**(3): p. 133-137.
2. Kim, H.Y., et al., *EPPS rescues hippocampus-dependent cognitive deficits in APP/PS1 mice by disaggregation of amyloid-beta oligomers and plaques*. Nature Communications, 2015. **6**.
3. Lee, S. and Y. Kim, *Anti-amyloidogenic Approach to Access Amyloid-(1-42) in Fmoc Solid-Phase Synthesis*. Bulletin of the Korean Chemical Society, 2015. **36**(8): p. 2147-2149.
4. Kim, H.V., et al., *Amelioration of Alzheimer's disease by neuroprotective effect of sulforaphane in animal model*. Amyloid-Journal of Protein Folding Disorders, 2013. **20**(1): p. 7-12.
5. Roman, I., et al., *Direct measurement of VDAC-actin interaction by surface plasmon resonance*. Biochimica Et Biophysica Acta-Biomembranes, 2006. **1758**(4): p. 479-486.
6. Kim, H.Y., et al., *Intracerebroventricular Injection of Amyloid-beta Peptides in Normal Mice to Acutely Induce Alzheimer-like Cognitive Deficits*. Jove-Journal of Visualized Experiments, 2016(109).
7. Jagtap, P.G., et al., *Structure-activity relationship study of tricyclic necroptosis inhibitors*. Journal of Medicinal Chemistry, 2007. **50**(8): p. 1886-1895.
8. Zhu, S., et al., *Necrostatin-1 ameliorates symptoms in R6/2 transgenic mouse model of Huntington's disease*. Cell Death & Disease, 2011. **2**.
9. Vitner, E.B., et al., *RIPK3 as a potential therapeutic target for Gaucher's disease*. Nature Medicine, 2014. **20**(2): p. 204-208.
10. Li, J.X., et al., *The RIP1/RIP3 Necrosome Forms a Functional Amyloid Signaling Complex Required for Programmed Necrosis*. Cell, 2012. **150**(2): p. 339-350.

Thank you for the submission of your revised manuscript to EMBO Molecular Medicine. We have now received the enclosed reports from the referees that were asked to re-assess it. As you will see, while referee 2 is now fully supportive, referee 1 remains critical. Before to make a final decision, we would like to ask you to provide a point-by-point response letter to this reviewer. We would not ask you to perform additional experiments at this point, but a rebuttal letter would help clarify the remaining concerns.

Please submit your revised manuscript within two weeks. I look forward to seeing a revised form of your manuscript as soon as possible.

***** Reviewer's comments *****

Referee #2 (Comments on Novelty/Model System):

Only one animal model has been investigated

Referee #2 (Remarks):

Unfortunately, in the revised version the criticism raised before has insufficiently been addressed. THE SPR data shown is even difficult to interpret for a specialist since it does not seem to follow any known binding models. Also, it is unclear what the model would be if Nec-1 binds Tau. Abeta and Tau would have to share some molecular properties that Nec-1 binds to both type of molecules as mentioned in the manuscript (Tau data not shown). In any case, it is problematic to analyze Abeta oligomers by SDS-PAGE/ PIGP analysis. Even more since ThT data are contradictory from what was concluded from the analysis in SDS gels. In addition, it would be crucial to show that Nec-1 affects phosphorylation of RIPK1 in vivo to strengthen the overall hypothesis.

Referee #3 (Remarks):

The authors clearly answers to my suggestions and the manuscript has been much improved. I have nothing to comment about revised manuscript.

2nd Revision - authors' response

11 October 2016

Based on your recommendation and the referee #2's comments, the following changes, additions and corrections have been made and are listed in point-by-point responses.

Referee #2 (Comments on Novelty/Model System):

Only one animal model has been investigated

→ Previously, referee #2 pointed out the same issue. We were told that an additional animal model was required and the main reason was to avoid any gene related artefacts in only one transgenic mouse model. Therefore, we performed additional behavioural experiments using a non-transgenic, A β -infused mouse model. In the current version of the manuscript, two animal models, with distinct Alzheimer-related culprits, were used for behavioural studies.

Referee #2 (Remarks):

Unfortunately, in the revised version the criticism raised before has insufficiently been addressed. The SPR data shown is even difficult to interpret for a specialist since it does not seem to follow any known binding models. Also, it is unclear what the model would be if Nec-1 binds Tau.

→ We apologized for insufficient explanation on the unusual shapes of SPR curves. Referee #2 pointed out that our data did not fit into commonly known bimolecular interactions in Biacore devices. However, we have observed the similar binding models, shown in this manuscript, while we measured bimolecular interactions between chemical analytes and heterogeneous misfolded proteins. Such results were recently published in Nature Communications [1], showing the same binding model between a small molecule and aggregated A β in Biacore. Therefore, in the revised manuscript, we clearly commented that the Biacore observation in this study is uncommon as we were assessing bimolecular interactions between a small molecule and misfolded proteins (line 22 of page 10 and line 16 of page 14).

Abeta and Tau would have to share some molecular properties that Nec-1 binds to both type of molecules as mentioned in the manuscript (Tau data not shown).

→ In contrast to the case of A β and its misfolded aggregates, there is no crystal structure of tau proteins or tangles. It was already challenging for us to predict binding pockets of Nec-1 in A β aggregates with limited information of A β structures. Tau is about 10 times larger than A β and it was impossible for us to predict how tau forms protein complexes. As it was beyond the scope of our study, we have to give up the molecular simulation between tau and Nec-1.

In any case, it is problematic to analyse A β oligomers by SDS-PAGE/ PICUP analysis. Even more since ThT data are contradictory from what was concluded from the analysis in SDS gels.

→ We are concerned about this comment by the referee #2 as SDS-PAGE PICUP is probably the most commonly used biochemical method to visualize the level of A β oligomers. The protocol papers by Gal Bitan and colleagues are cited by numerous amyloid-related reports for the reasons [2, 3]. Our laboratory also has published numbers of research articles using the PICUP method to show alteration of A β oligomers.

It is often misinterpreted when SDS-PAGE PICUP and ThT assays were performed together. The former indicated the levels of oligomers and fibrils, while the latter measured the levels of β -sheet-rich proto-fibrils and fibrils. Many research teams including us previously reported that analyses of molecules regulating A β aggregations could show different outcomes in SDS-PAGE PICUP and ThT assays as these methods are analyzing different states of A β aggregates. Despite such possibilities, our findings indicated that Nec-1 inhibited the formations of A β oligomers and fibrils in SDS-PAGE PICUP and that Nec-1 prevented the formation of β -sheet-rich proto-fibrils and fibrils in ThT assays. Therefore, we believed our findings are similar in both analyses.

In addition, it would be crucial to show that Nec-1 affects phosphorylation of RIPK1 in vivo to strengthen the overall hypothesis.

→ During our previous revision, we tried to examine the phosphorylation of RIPK1; however, an antibody against murine phosphorylated RIPK1 was not currently available. Alternatively, we conducted two different methods; 1) we performed co-immunoprecipitation using an antibody against RIPK1 followed by phosphor-serine antibody from brain lysates of APP/PS1 mice model, and 2) we used an antibody against human phosphorylated RIPK1 to expect cross-reactivity with murine phosphorylated RIPK1. Unfortunately, phosphorylated RIPK1 was not detectable by both methods. Although it was not possible to directly show Nec-1 affecting the phosphorylation of RIPK1, we believe that our result of reduced level of phosphorylated RIPK3 is sufficient to provide evidence that Nec-1 serves its anti-necroptotic role in the brain of APP/PS1 mice.

Reference

1. Kim, H.Y., et al., *EPPS rescues hippocampus-dependent cognitive deficits in APP/PS1 mice by disaggregation of amyloid-beta oligomers and plaques*. Nature Communications, 2015. **6**.
2. Bitan, G., A. Lomakin, and D.B. Teplow, *Amyloid beta-protein oligomerization - Prenucleation interactions revealed by photo-induced cross-linking of unmodified proteins*. Journal of Biological Chemistry, 2001. **276**(37): p. 35176-35184.
3. Rahimi, F., P. Maiti, and G. Bitan, *Photo-induced cross-linking of unmodified proteins (PICUP) applied to amyloidogenic peptides*. J Vis Exp, 2009(23).

Corresponding Author Name: YoungSoo Kim

Manuscript Number: EMM-2016-06566-V2